# Activation of TrkB signaling mitigates cerebellar anomalies caused by *Rbm4-Bdnf* deficiency

Yu-Young Tsai[1,4], Chiu-Lun Shen[1,2], Dhananjaya D[1], Ching-Yen Tsai[3] & Woan-Yuh Tarn [1✉]

A molecular and functional link between neurotrophin signaling and cerebellar foliation is lacking. Here we show that constitutive knockout of two homologous genes encoding the RNA binding protein RBM4 results in foliation defects at cerebellar lobules VI–VII and delayed motor learning in mice. Moreover, the features of *Rbm4* double knockout (dKO), including impaired differentiation of cerebellar granule cells and dendritic arborization of Purkinje cells, are reminiscent of neurotrophin deficiency. Loss of RBM4 indeed reduced brain-derived neurotrophic factor (BDNF). RBM4 promoted the expression of BDNF and full-length TrkB, implicating RBM4 in efficient BDNF-TrkB signaling. Finally, prenatal supplementation with 7,8-dihydroxyflavone, a TrkB agonist, restored granule cell differentiation, Purkinje cell dendritic complexity and foliation—the intercrural fissure in particular—in the neonatal cerebellum of *Rbm4*dKO mice, which also showed improved motor learning in adulthood. This study provides evidence that prenatal activation of TrkB signaling ameliorates cerebellar malformation caused by BDNF deficiency.

[1] Institute of Biomedical Sciences, Academia Sinica, Taipei, Taiwan. [2] Institute of Molecular Medicine, College of Medicine, National Taiwan University, Taipei, Taiwan. [3] Institute of Molecular Biology, Academia Sinica, Taipei, Taiwan. [4] Present address: Columbia University in the City of New York, New York, US. ✉email: wtarn@ibms.sinica.edu.tw

The RNA binding protein RBM4 modulates precursor mRNA splicing and mRNA translation. Depletion of RBM4 impairs the differentiation of various cell types, including neuronal cells[1–6]. RBM4 promotes neuronal differentiation by modulating alternative splicing of the cell-fate determinant Numb and the glycolytic enzyme pyruvate kinase M[4,5]. Moreover, an *in-utero* experiment demonstrated that RBM4 participates in radial migration of newborn neurons in the developing cerebral cortex by promoting the expression of full-length Dab1[6], a regulator of Reelin signaling that controls neuronal positioning. However, whether RBM4 is essential for brain development has not been evaluated.

In this study, we observed that constitutive knockout of *Rbm4* genes caused cerebellar vermis hypoplasia in specific lobules. Anatomically, the mammalian cerebellum consists of ten vermian lobules, and the cortex consists of three layers, i.e., the Purkinje cell (PC), granule cell (GC), and molecular layers[7]. In the early stage of cerebellar development, GC precursors (GCPs) migrate tangentially over the cerebellar primordium to the external granule layer (EGL). GCPs then continue to proliferate and differentiate in the EGL, and subsequently postmitotic GCs migrate radially to the internal granule layer (IGL) through the PC layer during the early postnatal period. PCs are aligned in the PC layer, with their complex dendritic arbors extending into the molecular layer. Functionally, cerebellar folia serve as an integrative relay station for organizing the sensory-motor circuits of the cerebellum[8]. Genetic studies in mouse models have revealed a number of transcription factors, chromatin-remodeling factors, and signaling factors involved in cerebellar morphogenesis, such as the engrailed family of transcription factors and the Sonic Hedgehog signaling pathway[9]. As expected, knockout of engrailed 2 or Sonic Hedgehog impairs foliation and cerebellar development[10,11].

In addition, several factors such as neurotrophins and thyroid hormone are essential for cerebellar development. Knockout of brain-derived neurotrophic factor (BDNF) or its receptor TrkB affects cerebellar development. *Bdnf* knockout results in increased apoptosis, delayed migration of GCs, stunted growth of PC dendrites, and the absence of the intercrural fissure (icf) that divides lobules VI and VII[12]. Constitutive knockout of *TrkB* moderately affects icf formation and reduces the dendritic complexity of PCs[13]. Ablation of TrkB in cerebellar precursors impairs synapse formation of inhibitory interneurons in the cerebellum[14]. Furthermore, disruption of BDNF release or signaling also impacts cerebellar development. For example, knockout of *CAPS2*, which encodes a secretory granule–associated protein involved in BDNF secretion, impairs GC migration and PC arborization[15]. The Vav family of guanine nucleotide exchange factors acts downstream of the BDNF-TrkB signaling[16]. Upon BDNF stimulation, Vav promotes dendritic spine growth and synaptic plasticity[17]. *Vav3* knockout compromises GC migration and PC dendrite development[18]. Maternal thyroid hormone deficiency likely causes foliation defects in offspring mice via BDNF downregulation[19,20], underscoring the importance of BDNF in cerebellar development.

Hypoplasia of vermian lobules VI–VII has been linked to autism spectrum disorder[21]. BDNF-deficient, *Trkb* mutant, and *Caps2* knockout mice exhibit defective icf formation and altered behaviors, including exploration and anxiety[15,22,23]. Therefore, further investigation is warranted concerning the molecular mechanisms underlying the relationship between cerebellar malformations and neurological disorders. Here, we found that *Rbm4* knockout mice exhibited cerebellar defects similar to what has been observed in *Bdnf*-deficient mice. We thus evaluated whether reactivation of TrkB signaling could ameliorate such defects.

## Results

**Abnormal cerebellar foliation in *Rbm4* knockout mice.** Mammalian genomes contain two *Rbm4* genes, namely *Rbm4a* and *Rbm4b*. These two *Rbm4* genes are highly similar with respect to exon/intron structure and are oriented in opposite directions on the same chromosome, of which the encoded proteins share 85% identity with each other. To explore the role of RBM4 in brain development, we had attempted to establish conditional *Rbm4a/ Rbm4b* double-knockout in the embryonic cortex. Therefore, using CRISPR-Cas9A$^{D10A}$-mediated gene editing, we inserted a loxP site in *Rbm4b* of previously established *Rbm4a* knockout mice[2] (Supplementary Fig. 1a, $Rbm4a^{-/-};Rbm4b^{f/f}$). However, *Emx1* promoter–driven Cre recombinase failed to excise the ~30-kb floxed segment encompassing both *Rbm4a* and *Rbm4b*, perhaps owing to insufficient or transient expression of Cre. Nevertheless, successful excision was achieved by Cre whose expression was driven by the strong *EIIa* promoter in the early mouse embryo (Fig. 1a and Supplementary Fig. 1a), resulting in constitutive homozygous *Rbm4a/Rbm4b* double-knockout ($Rbm4a^{-/-};Rbm4b^{-/-}$, abbreviated as *Rbm4*dKO). The *Rbm4*dKO genotype was confirmed by PCR (Fig. 1b). This double knockout completely abolished *Rbm4a/b* expression at both the mRNA and protein levels (Fig. 1b). *Rbm4*dKO mice were viable and fertile, with only a slightly reduced growth rate (Supplementary Fig. 1b). *Rbm4a/b* (hereafter abbreviated as *Rbm4*) knockout did not alter the gross morphology of the brain (Supplementary Fig. 1c). Hematoxylin-eosin (HE) staining of a coronal section of the P10 (postnatal day 10) *Rbm4*dKO cerebrum revealed apparently normal cortical organization and laminar structure as well as hippocampal morphology and size (Supplementary Fig. 1d, e). On the other hand, the *Rbm4*dKO cerebellum, albeit grossly normal in morphology, foliation pattern, lobule number and midsagittal vermal area at P30 (Supplementary Fig. 1f–i), developed slightly slower than wild type and had delayed foliation during the early neonatal period (Fig. 1c, d and Supplementary Fig. 1j). In particular, the boundary between vermian lobules VI and VII, i.e., the icf, was less prominent in the *Rbm4*dKO cerebellum compared with wild type (Fig. 1c, e). Because *Rbm4* knockout caused cerebellar anomaly, we examined whether RBM4 is expressed in the cerebellum. Immunoblotting revealed continuous RBM4 expression in the embryonic and postnatal cerebellum (Supplementary Fig. 1k). Using immunohistochemical staining, we observed that RBM4 was likely present in both PCs and GCs of the P10 cerebellum and its robust signal in the molecular layer may represent PC dendrites (Fig. 1f). No visible RBM4 immunoreactivity in the *Rbm4*-knockout cerebellum indicated complete knockout (Fig. 1f). These results revealed that RBM4 is expressed in the cerebellum and its knockout affects vermis icf formation.

**_Rbm4_ knockout affects motor learning as well as exploratory and anxiety-like behavior.** Given that abnormal cerebellar morphology is associated with motor learning deficits, we first performed rotarod and open-field analyses. For the rotarod analysis, mice were pretrained to ambulate on the rotarod at a constant speed before the test over four sessions (Supplementary Fig. 2). During the actual test, the rotational speed gradually increased. We found that *Rbm4*dKO mice remained on the accelerating rotarod for a shorter period compared with wild-type mice in the first two trials but attained comparable performance level of wild type by the third trial, suggesting that *Rbm4* ablation results in a transient delay in motor learning (Fig. 2a). In the open-field analysis, *Rbm4*dKO mice traveled greater distances over the habituation period and spent significantly more time in the center as compared with wild type, indicative of their

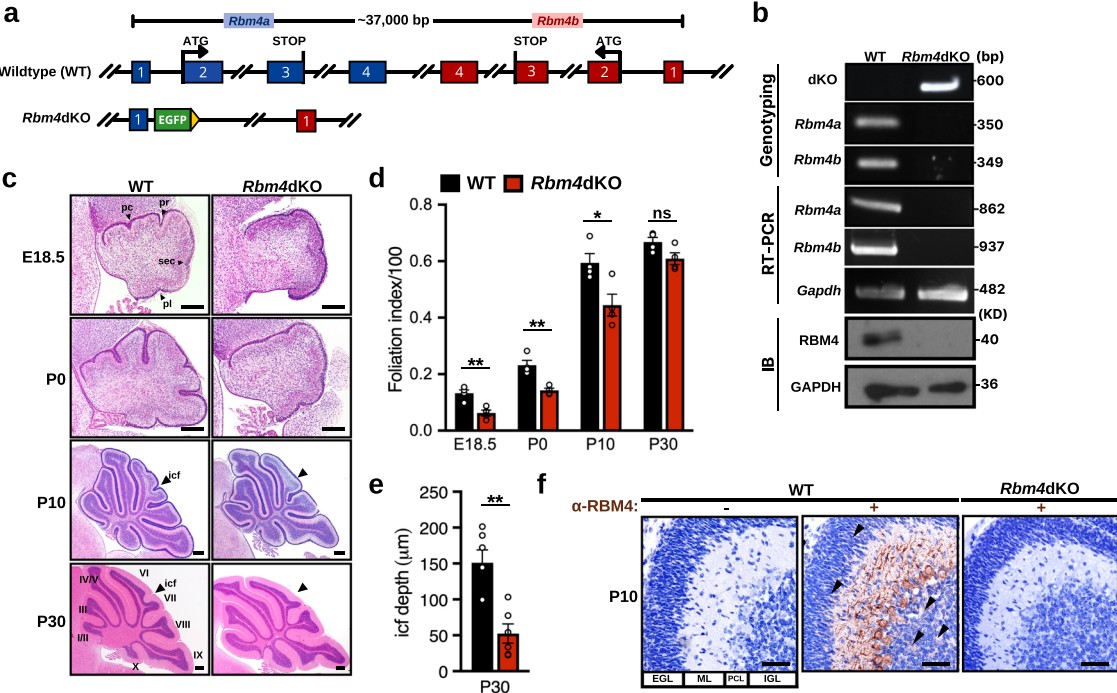

**Fig. 1 Rbm4 knockout disrupts cerebellar icf formation between vermian lobules VI and VII. a** Scheme shows the genomic organization of *Rbm4a/b* in wild-type (WT) and *Rbm4*dKO mice. A more detailed scheme is presented in Supplementary Fig. 1a. **b** Analysis of WT and *Rbm4*dKO mice: genotyping (upper panel), RT-PCR (middle), and immunoblotting (IB, bottom) of E13.5 whole-brain lysates. Primers used for PCR and RT-PCR are detailed in Supplementary Table 1 and Supplementary Fig. 1a. **c** Representative images of HE staining of the sagittal vermis in WT and *Rbm4*dKO mice at E18.5, P0, P10, and P30. Roman numerals indicate relevant lobules of the vermis. Arrowheads indicate the presence or absence of an icf. **d** Foliation index, computed as [1 – (convex length/EGL length) ×100], of the cerebellar vermal surface for the indicated age as in (**c**) (*N* = 4 per genotype). **e** Average depth of icf at P30 (*N* = 6 per genotype). **f** Immunohistochemistry against RBM4 in the P10 cerebellum (developed with the chromogen diaminobenzidine and counterstained with hematoxylin) of WT and *Rbm4*dKO mice. RBM4 was highly expressed in the molecular layer, PCL, and detectable in both EGL and IGL of wild type (arrowheads). pc, pre-culminate fissure; pr, primary fissure; sec, secondary fissure; pl, posterolateral fissure; icf, intercrural fissure; EGL, external granule layer; ML, molecular layer; PCL, Purkinje cell layer; IGL, internal granule layer. Scale bars, 200 μm (**c**); 50 μm (**f**). *P*-values were determined with the Student's *t* test: \**P* < 0.05, \*\**P* < 0.01, \*\*\**P* < 0.001; ns, difference not statistically significant. Error bars represent the standard error of the mean.

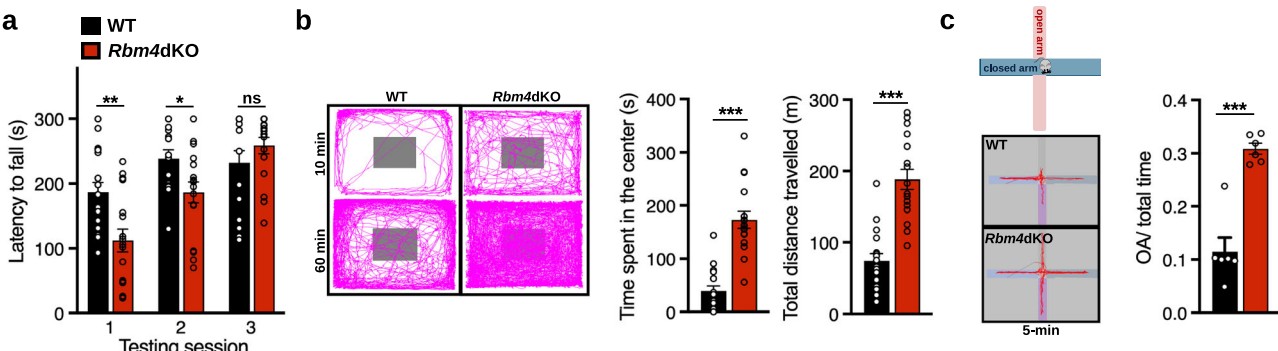

**Fig. 2 Rbm4 knockout affects motor learning and exploratory behavior. a** For rotarod analysis, each WT and *Rbm4*dKO mouse (two-month-old) was trained on the rod for 60 s at low constant speed (Supplementary Fig. 2) before proceeding to the three tests with an accelerating rotational speed from 4 to 40 rpm over a period of 300 s. Bar graph shows the average latency to fall (s) from the rotarod (*N* = 17 per genotype). **b** For the open-field test, the movement of WT and *Rbm4*dKO mice was recorded over the indicated habituation period (left). Bar graphs show average time spent in the center (middle) and the total distance traveled during the 1-h exploration (right; *N* = 17 per genotype). **c** Schematic illustration of the EPM analysis (upper left). Representative mouse movement tracks during the EPM test over the 5-min habituation period (bottom left). Bar graph (right) shows the relative duration spent in the open arm during the 5-min habituation (*N* = 6 per genotype). *P*-values and error bars are the same as in Fig. 1.

enhanced exploratory and reduced anxiety-like behaviors (Fig. 2b). To further assess these behaviors, we performed the elevated plus maze (EPM) test, which is sensitive to anxiety-related behaviors. As compared with wild-type mice, *Rbm4*dKO mice spent significantly more time exploring the open arms

(Fig. 2c), consistent with the behavioral pattern observed in the open-field paradigm. Therefore, *Rbm4* knockout likely reduced anxiety-like behaviors. However, in light of global knockout of *Rbm4*, the concern that abnormal behaviors resulted from *Rbm4* deficiency in other brain regions certainly remains.

**RBM4 regulates the cell cycle and migration of GCs during cerebellar development**. The aforementioned data indicated that *Rbm4* knockout caused a transient delay in cerebellar foliation and disrupted timely icf formation (Fig. 1). Cerebellar foliation proceeds with GCP proliferation in the EGL and subsequent GC migration to the IGL. Using hematoxylin staining, we observed that the EGL thickness of the P10 *Rbm4*dKO cerebellum was significantly greater than that of wild type (Supplementary Fig. 3a). This result prompted us to examine whether the proliferation of GCPs and/or kinetics of GC migration are impaired in *Rbm4*dKO mice. As reported[24], pulse-chase labeling with bromodeoxyuridine (BrdU) revealed that labeled GCPs were mainly distributed in the EGL at 2-h post-injection and subsequently migrated into the IGL (Supplementary Fig. 3b). After a 48-h chase period, co-staining of the BrdU-labeled cerebellum with the cell-proliferation marker Ki67 revealed that the Ki67-positive layer was thicker in the *Rbm4*dKO cerebellum than in the wild type (Fig. 3a, b). The percentage of GCs that had recently exited the cell cycle (Ki67$^-$/BrdU$^+$) and localized to the inner EGL (iEGL) was accordingly reduced in the *Rbm4*dKO cerebellum (Fig. 3a, lobule VI), indicating that RBM4 depletion may prolong the retention of GCPs in the proliferative phase. A similar defect was observed in all other lobules (Fig. 3c). Co-staining with the GC pan marker Pax6 revealed that *Rbm4* knockout reduced the proportion of BrdU-labeled cells migrating out of the EGL and into the cortex, suggesting that impaired cell cycle exit consequently caused a delay in migration (Fig. 3d, lobule VI). This phenomenon was also consistent among all lobules examined (Fig. 3e). Collectively, these results indicated that *Rbm4* knockout prolonged the cell cycle of GCPs within the outer EGL and hence delayed differentiation of new GCs (Fig. 3f). Accordingly, immunohistochemistry using a mature neuronal marker NeuN revealed a significant reduction of mature GCs in the cerebellar cortex (Fig. 3g). In addition, *Rbm4* knockout increased the number of apoptotic cells at P0, as revealed by immunostaining against cleaved caspase-3 (Supplementary Fig. 3c). Together, these results implied that RBM4 regulates cell-cycle and subsequent differentiation, and survival of GCs in the developing cerebellum.

**RBM4 is critical for dendritic arborization of PCs**. Since RBM4 was found to be abundant in PCs (Fig. 1f), we examined whether *Rbm4* knockout affected PC development. Immunofluorescence staining for the calcium-binding protein calbindin revealed that P8 *Rbm4*dKO PCs exhibited markedly reduced dendritic arborization (Fig. 3h), as evidenced by a ~40% reduction in the length of primary dendrites (Fig. 3i). Nevertheless, the number of PCs was not affected by *Rbm4* knockout (Fig. 3j). Finally, both anti-calbindin staining and Golgi staining of P30 cerebellum revealed no significant difference in PC dendritic complexity between the wild-type and *Rbm4*dKO (Supplementary Fig. 3d). This result was not completely unexpected, as the gross morphology of young adult *Rbm4*dKO cerebellum, except for the icf malformation, was normal. Therefore, RBM4 is particularly critical for early dendrite development of PCs.

**Loss of RBM4 reduces BDNF expression and alters the TrkB isoform ratio**. The results shown thus far revealed several prominent features of the *Rbm4*dKO cerebellum, including defective icf formation, delayed GC differentiation, and impaired PC dendritic arborization (Figs. 1 and 3), which were remarkably similar to the mutants with defective BDNF signaling[12,15,18,25]. Therefore, we evaluated BDNF expression in *Rbm4*dKO mice. *Bdnf* mRNA was continuously expressed in the embryonic brain and developing cerebellum, and *Rbm4* knockout reduced *Bdnf*

mRNA level by 40–60% at all time points examined except for embryonic day 13.5 (E13.5; Fig. 4a). BDNF protein was barely detectable at E13.5 (Fig. 4b), although its mRNA level was comparable to that measured at other time points. Nevertheless, mature BDNF was detectable in the E18.5 whole brain and the developing cerebellum. *Rbm4* knockout diminished BDNF protein level by 60–90% (Fig. 4b). These findings suggested a possible role for RBM4 in regulating BDNF expression probably at the transcriptional level; its underlying mechanism remains to be investigated. Given that BDNF is indispensable for the survival and migration of GCs and dendrite development of PCs[12,24], its reduction may thus cause malformation of the *Rbm4*dKO cerebellum.

While examining TrkB activation, we had fortuitously found that a truncated TrkB isoform (TrkB.T1), an alternatively spliced and kinase-domain-lacking TrkB isoform, was increased in *Rbm4*dKO cerebellum. Reverse transcription–coupled PCR (RT-PCR) analysis revealed a splicing shift from full-length *Ntrk2* (*Ntrk2*-fl, encoding TrkB) to a truncated *Ntrk2*-t1 isoform (encoding TrkB.T1) in the *Rbm4*dKO cerebellum (Fig. 4c). Accordingly, immunoblotting revealed an increase in TrkB.T1 in *Rbm4*dKO cerebellar lysates (Fig. 4d). Therefore, it was conceivable that *Rbm4* knockout could impair BDNF-TrkB signaling at two different levels.

**RBM4 regulates the expression of both BDNF and TrkB**. To better define the role of RBM4 in BDNF and TrkB gene expression, we performed in vitro experiments using GCs isolated from wild-type mice. Immunocytochemical staining for Pax6 revealed that the isolated GCs yielded a purity of 90–95% (Fig. 5a). Upon transfection of short interfering RNA targeting *Rbm4*[5], both *Rbm4* transcripts were significantly downregulated (Fig. 5b). *Rbm4* knockdown reduced *Bdnf* mRNA expression by ~70% and *Ntrk2*-fl splicing by up to fivefold (Fig. 5b, gel images for RT-PCR for *Bdnf* and *Ntrk2*, and bar graph for quantitative RT-PCR of *Bdnf*), consistent with observations made with the *Rbm4*dKO cerebellum (Fig. 4). Next, a recombinant adeno-associated virus (AAV) expressing mouse RBM4 was generated to evaluate whether RBM4 promotes *Bdnf* expression and the *Ntrk2* isoform switch. AAV-mediated overexpression of RBM4 promoted *Bdnf* expression and *Ntrk2* splicing towards the full-length isoform (Fig. 5c), indicating that RBM4 is critical for the expression of both BDNF and full-length TrkB. Furthermore, we assessed whether exogenous RBM4 could restore *Bdnf* and *Ntrk2*-fl expression in *Rbm4*dKO GCs. Indeed, transduction with AAV-RBM4 restored *Bdnf* mRNA expression and the splicing switch from *Ntrk*-t1 to *Ntrk*-fl (Fig. 5d). Both immunoblotting and immunocytochemical staining confirmed an increase of BDNF expression in *Rbm4*dKO GCs upon AAV-RBM4 transduction (Fig. 5d, e). Importantly, RBM4 overexpression enhanced phosphorylation of TrkB (Y816) in *Rbm4*dKO GCs (Fig. 5e), supporting the expression and activation of full-length TrkB. These in vitro results indicated that RBM4 is critical for efficient BDNF-TrkB signaling.

**A small-molecule TrkB agonist partially rescues the developmental defects of the *Rbm4*dKO cerebellum**. *Rbm4* knockout disrupted the BDNF-TrkB pathway in the developing cerebellum (Figs. 4 and 5). To correct the developmental anomalies of the *Rbm4*dKO cerebellum, we thought to activate the remaining full-length TrkB with a small-molecule TrkB agonist. 7,8-DHF is a potent TrkB agonist that promotes its autophosphorylation and consequent downstream signaling, and this agonistic action is enabled by its ability to penetrate the placenta and blood-brain barrier[26]. Therefore, *Rbm4*dKO embryos were prenatally

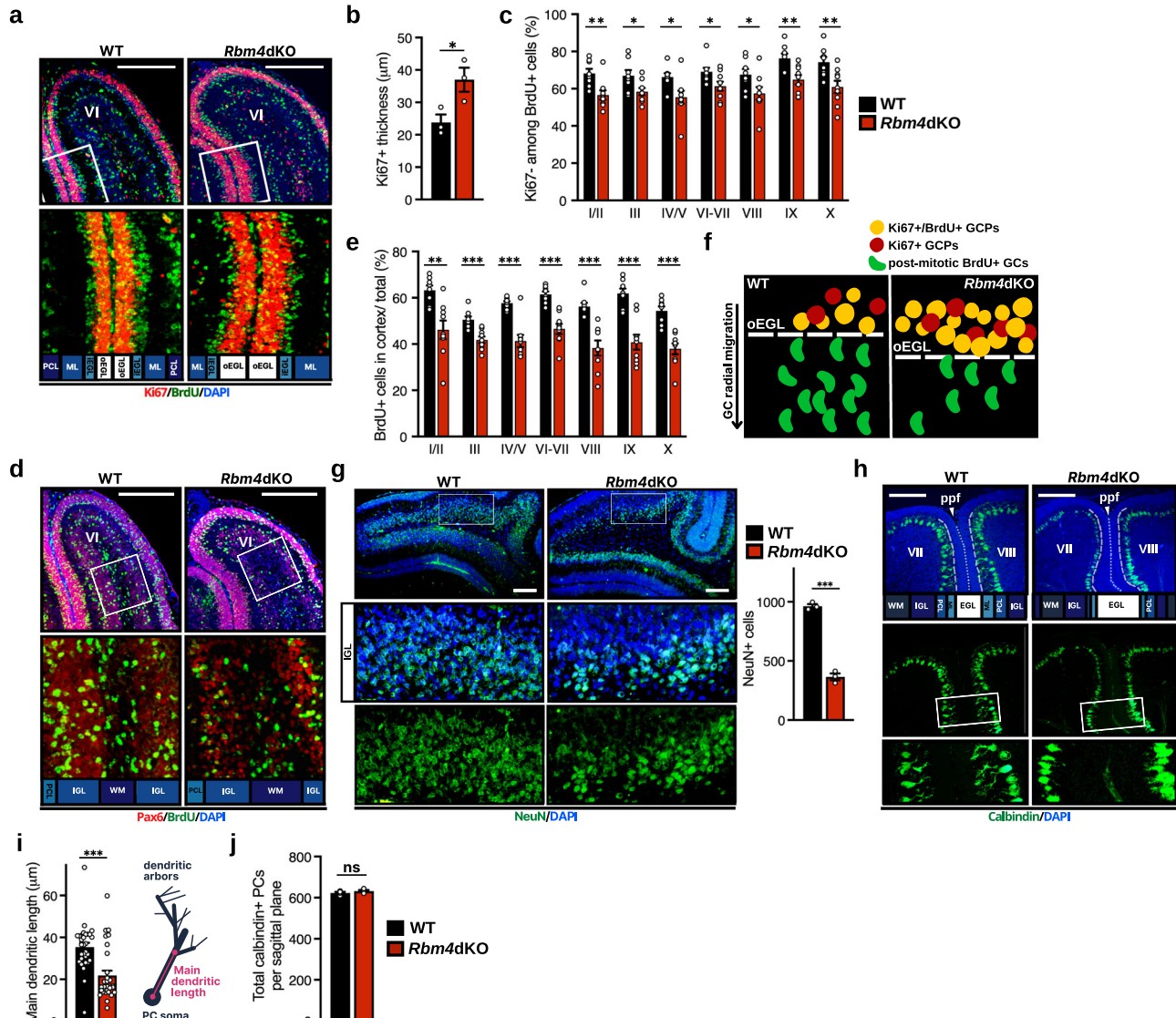

**Fig. 3 *Rbm4* knockout delays GC differentiation and impairs PC dendritogenesis. a** Intraperitoneal BrdU injection was performed at P6, and the brain was harvested 48 h post-injection. Cerebellar sections were co-immunostained for BrdU and Ki67; nuclei were counterstained with DAPI (upper panels). Representative images are shown above; the boxed regions are magnified in the lower panels. Layers of the cerebellar cortex are indicated below the images. Abbreviations are as in Fig. 1. **b** Quantification of the width of the Ki67$^+$ layer of lobule VI. Average thickness was sampled from three stainings per genotype. **c** Quantification of the percentage of Ki67$^-$ among BrdU$^+$ cells in the EGL of the indicated folium. Sections were averaged from WT: $N = 8$, *Rbm4*dKO: $N = 9$. **d** Cerebellar sections were co-immunostained for BrdU and Pax6. Representative images are shown as in (**a**). **e** The relative proportion of BrdU$^+$ cells that migrated into the cerebellar cortex (i.e., exited from EGL) was calculated in the indicated folium. Sampling was as in (**c**). **f** Schematic illustration of proliferation and migration of GCs in the WT or *Rbm4*dKO cerebellum. Loss of *Rbm4* resulted in retention of GCP in the proliferative phase and defective radial migration. **g** Representative immunostaining against NeuN in each genotype is shown; the magnified region (white box) is shown below in merged and single channel, respectively. Bar graph (right) shows the quantification of NeuN+ cells ($N = 3$ per genotype). Quantification of NeuN + cells was performed in the IGL-PCL cortical zone of lobules VI–VII. **h** Representative immunofluorescence staining for calbindin at P8 (top subpanels). Layers of the cerebellar cortex are labeled below the images, and boxed regions are magnified in the lower panels. **i** Average length of the main dendrite was measured from the midpoint of the soma to the first major branch point, or as far as it was visibly traceable if in the absence of which (right). 30 PCs were sampled from three mice per genotype. **j** Average total number of calbindin$^+$ cells ($N = 4$ per genotype). Scale bars, 200 μm. *P*-values and error bars are the same as in Fig. 1.

supplemented with 7,8-DHF via daily injection into pregnant *Rbm4*dKO females starting at E15.5 and until birth (Fig. 6a, mice born from mock- and 7,8-DHF-treated mothers were termed *Rbm4*dKO$^{Mo}$ and *Rbm4*dKO$^{DHF}$, respectively). Immunoblotting revealed a considerable increase in the level of phosphorylated TrkB in P0 *Rbm4*dKO$^{DHF}$ (Fig. 6b; Supplementary Fig. 4a, cerebrum and cerebellum), indicating the transient effect of prenatal 7,8-DHF supplementation on TrkB activation. Activated TrkB signaling upregulates the expression and secretion of BDNF via a

positive autoregulatory loop[27,28]. Accordingly, the level of BDNF was increased in *Rbm4*dKO$^{DHF}$ but not *Rbm4*dKO$^{Mo}$ mice (Fig. 6b). Therefore, prenatal treatment with 7,8-DHF was able to activate TrkB signaling in the developing cerebellum.

HE staining revealed that prenatal supplementation with 7,8-DHF restored foliation in the P0 *Rbm4*dKO$^{DHF}$ cerebellum (Fig. 6c, d, left). More strikingly, *Rbm4*dKO$^{DHF}$ displayed well developed icf at P30 (Fig. 6c, d, right). At the level of cytoarchitecture, the thickness of Ki67-labeled EGL was

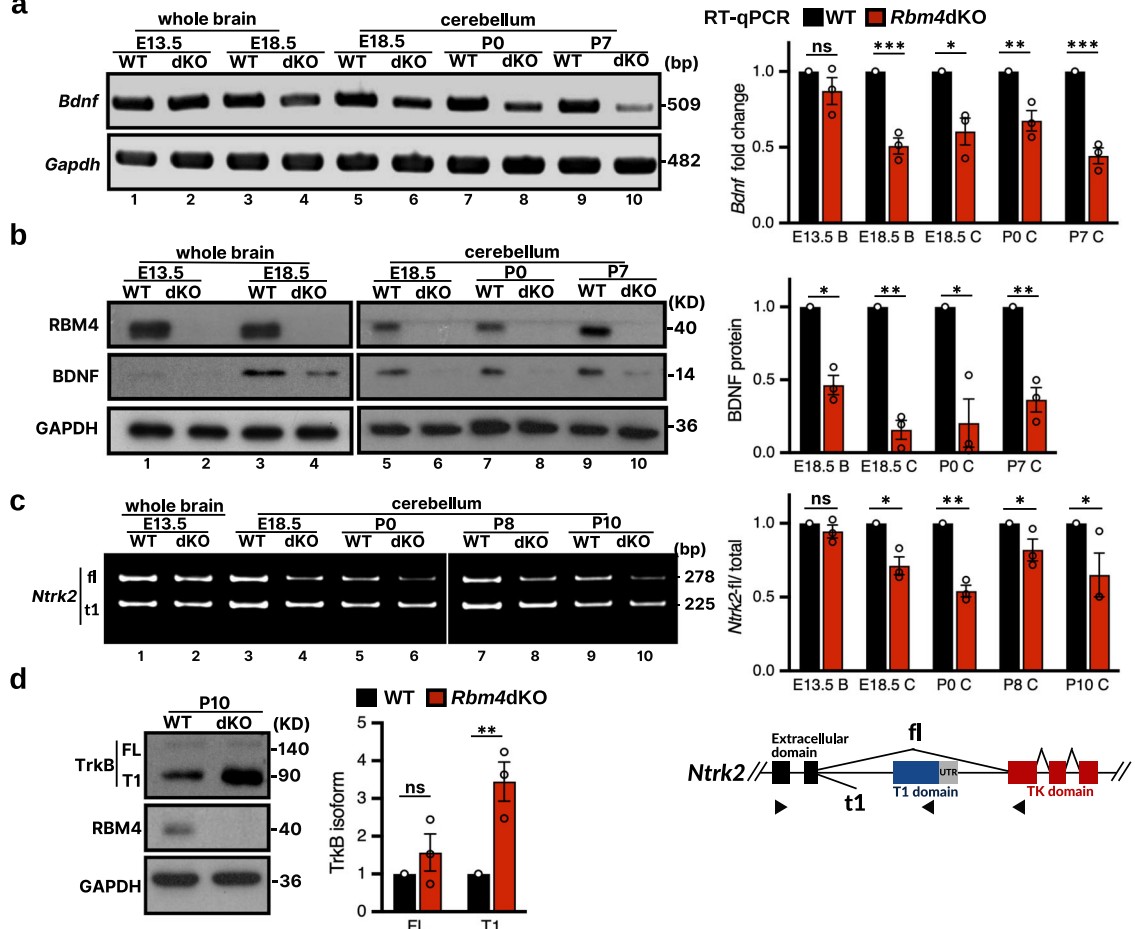

**Fig. 4 Rbm4 knockout reduces BDNF expression and alters the TrkB isoform ratio. a** *Bdnf* mRNA was examined in lysates from WT and *Rbm4*dKO whole brain or cerebellum at the indicated age by PCR (left) and quantitative RT-PCR (RT-qPCR, right; *N* = 3 per genotype). "B" and "C" in X-axis labels of the bar graph denote lysates of the brain and cerebellum, respectively. **b** Immunoblotting for BDNF, RBM4 and GAPDH was performed at the indicated ages. The bar graph (right) shows the relative level of BDNF after normalization to GAPDH (WT was set to 1; *N* = 3 per genotype). **c** Expression of *Ntrk2* isoforms in WT and *Rbm4*dKO cerebellum at the indicated postnatal days was assessed by RT-PCR using primers as depicted (black arrowheads in the scheme below). The relative *Ntrk2*-fl isoform ratio is shown to the right (*N* = 3 per genotype). The scheme below shows the gene structure and alternative splicing of *Ntrk2*. **d** Immunoblotting of RBM4, TrkB, and GAPDH from lysates of P10 cerebellum. Quantification of each TrkB isoform was normalized to GAPDH level (WT was set to 1; *N* = 3 per genotype). *P*-values and error bars are the same as in Fig. 1.

significantly reduced to the wild-type level in the P0 *Rbm4*dKO$^{DHF}$ cerebellum, suggesting a partial rescue of the *Rbm4*dKO neonatal cerebellum from delayed kinetics of GC proliferation (Fig. 6e, f). Accordingly, the thickness of the Pax6$^+$ EGL in *Rbm4*dKO$^{DHF}$ was also restored to wild-type level (Supplementary Fig. 4b). Treatment with 7,8-DHF also reduced the number of apoptotic cells in the cerebellar cortex of *Rbm4*dKO mice (Supplementary Fig. 4c). Moreover, in the anterior region of the vermis, the compact distribution of PCs—which originate from the subisthmal cerebellar primordium—was recovered in the P0 *Rbm4*dKO$^{DHF}$ cerebellum (Supplementary Fig. 4d). The dendritic complexity of *Rbm4*dKO$^{DHF}$ PCs at P8 was indeed restored to resemble that of wild type, indicating that treatment with 7,8-DHF also improved PC dendritogenesis (Fig. 6g, h). Therefore, prenatal treatment with 7,8-DHF reactivated TrkB signaling during early stages of cerebellar development, which was sufficient to restore GCP cell-cycle exit, PC arborization, and importantly, icf formation in *Rbm4*dKO mice (Fig. 6i). Wild-type mice were analogously treated (respectively termed WT$^{DHF}$ and WT$^{Mo}$; Supplementary Fig. 4a, e–h). However, prenatal 7,8-DHF supplementation did not further increase phospho-TrkB in wild-type mice

(Supplementary Fig. 4a, WT), likely due to their near-saturated TrkB activity. No significant difference between mock- and 7,8-DHF-treated wild-type mice was observed in all morphologies examined, including apoptotic events, GC layer thickness, PC dendritic complexity, and foliation (Supplementary Fig. 4e–h, respectively).

We wondered whether neonatal supplementation suffices in alleviating *Rbm4*dKO-induced cerebellar defects. To test this, a separate intervention approach was performed, which involved subcutaneous injection of 7,8-DHF into newborn pups from P5 to P30. Such regimen, albeit increasing the level of phosphorylated TrkB and BDNF, in *Rbm4*dKO, did not significantly improve their foliation defect at P30 (Supplementary Fig. 4i-k). The collective results above highlighted the therapeutic benefit conferred by prenatal compensation of BDNF-TrkB signaling for cerebellar icf formation in the mutant mice.

**Prenatal administration of 7,8-DHF restores motor learning.** Our results thus far indicated that RBM4 modulates cerebellar development through the BDNF-TrkB signaling pathway, and prenatal reactivation of TrkB was sufficient to rescue the cerebellar

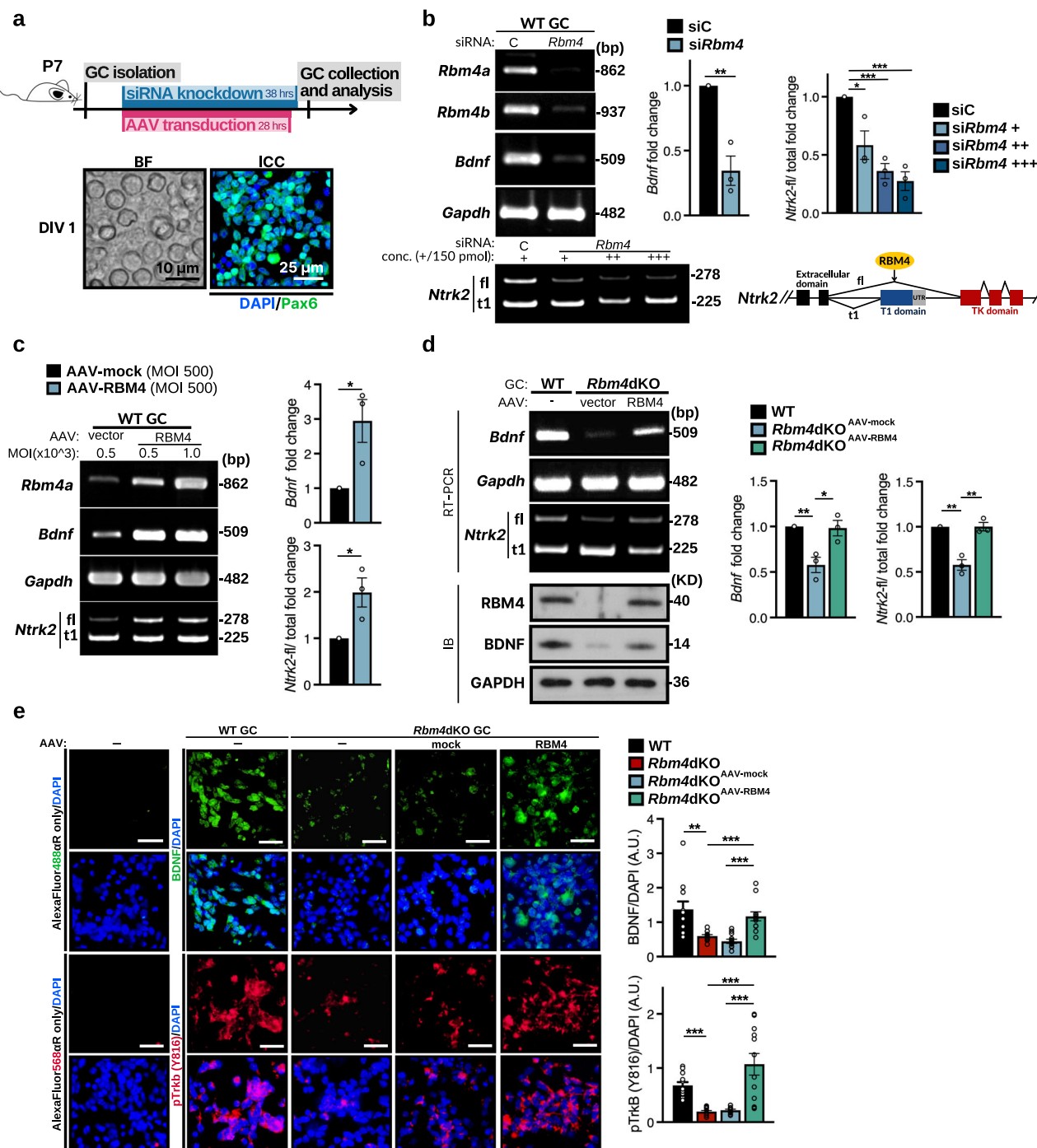

**Fig. 5 RBM4 regulates *Bdnf* and *Ntrk2* expression in cerebellar GCs. a** Schematic of the in vitro experiments using cerebellar GC(P)s that were isolated from P7 WT or *Rbm4*dKO mice. GCs were cultured one day in vitro (DIV) for siRNA-mediated *Rbm4* knockdown or transduction with AAV encoding RBM4. Immunocytochemistry for Pax6 confirmed GC identity and purity. **b** *Rbm4* was knocked down in WT GCs. Expression of *Rbm4*, *Bdnf*, and *Ntrk2* was evaluated by RT-PCR (left). *Gapdh* was used as a PCR control. Bar graphs show the relative level of *Bdnf* mRNA determined by RT-qPCR (middle) and the relative *Ntrk2*-fl isoform ratio (right). For both, WT was set to 1 (*N* = 3 independent experiments per group). The scheme is shown as in Fig. 4c (lower right). RBM4 promotes the expression of *Ntrk2*-fl via alternative splicing. **c** WT GCs were transduced with AAV-mock or AAV-RBM4 at the indicated multiplicity of infection (MOI). RT-qPCR and quantification were performed as in (**b**) (right; *N* = 3 independent experiments per group). **d** *Rbm4*dKO GCs were transduced with AAV-mock or AAV-RBM4. Gels show results from RT-PCR and immunoblotting analyses. The bar graphs show results from RT-qPCR of *Bdnf* and RT-PCR of relative *Ntrk2*-fl isoform ratio, as in (**b**) and (**c**). **e** Representative immunocytochemical staining against BDNF or phospho-TrkB (Y816) in isolated GCs upon transduction with AAV-mock or AAV-RBM4; nuclei were counterstained with DAPI. The left-most panels show immunofluorescence using the respective secondary antibody only as a control. Bar graphs (right) show fluorescence signals averaged from 12 randomly selected fields of 100 × 100 μm each. Scale bars, 25 μm (**e**). *P*-values and error bars are the same as in Fig. 1.

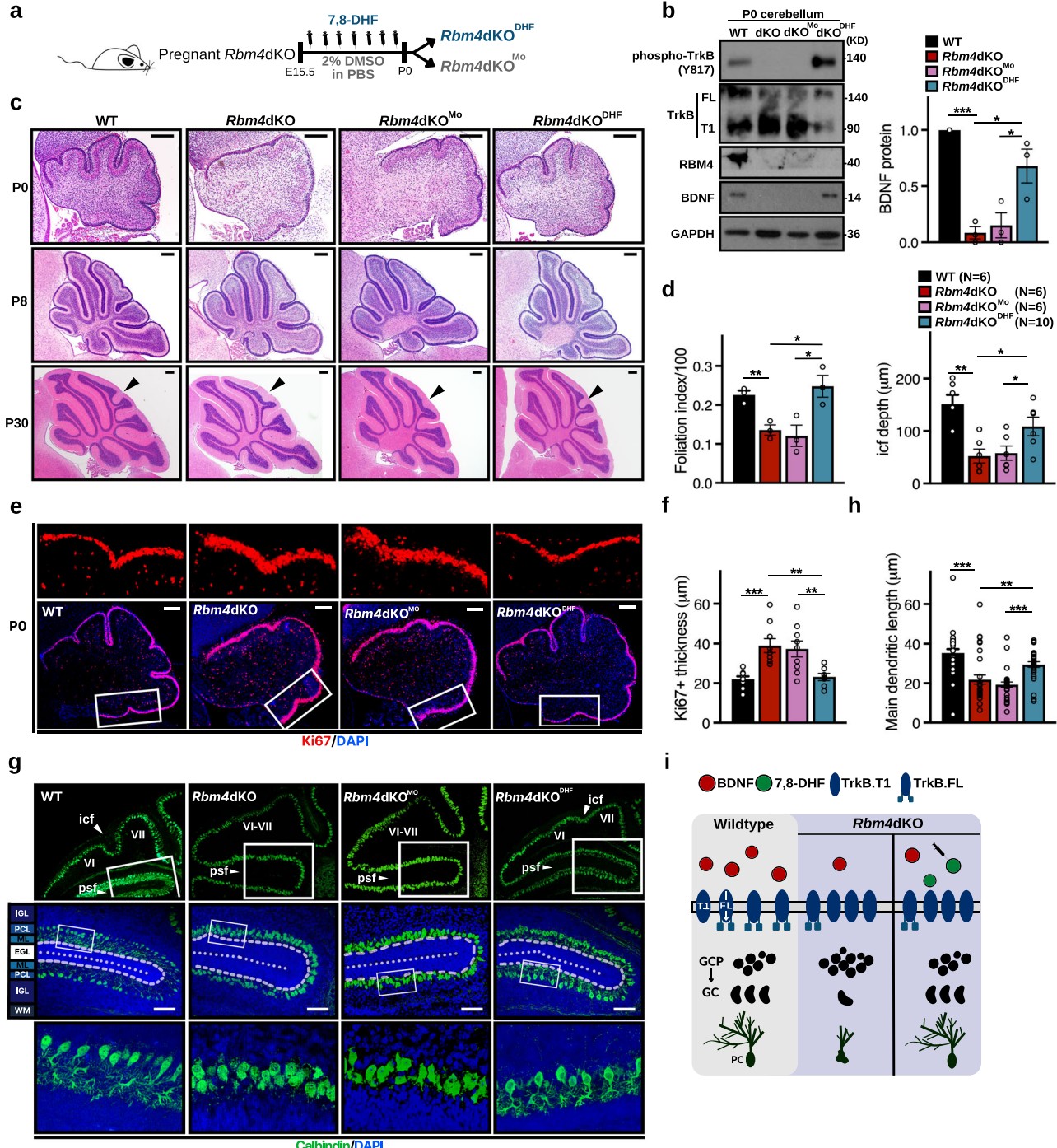

**Fig. 6 A small-molecule TrkB agonist reverses cerebellar malformation in *Rbm4*dKO mice. a** Experimental design for prenatal 7,8-DHF administration. Pregnant females were subcutaneously injected with 7,8-DHF (5 mg/kg) daily starting at E15.5 and until birth (P0). Newborn *Rbm4*dKO pups were termed *Rbm4*dKO^Mo and *Rbm4*dKO^DHF, respectively. **b** Immunoblotting of cerebellar lysates (WT, *Rbm4*dKO, *Rbm4*dKO^Mo and *Rbm4*dKO^DHF) collected at P0 using antibodies against phospho-TrkB (Y817), TrkB, BDNF, RBM4, and GAPDH. Quantification of BDNF is shown to the right (*N* = 3 per group). **c** Representative images of HE staining of the sagittal vermis for the indicated genotype/treatment/age. Arrowheads indicate the presence or absence of an icf at P30. **d** Bar graphs show the foliation index of the P0 cerebellum (left; *N* = 3 per group) and average icf depth at P30 (right; *N* indicated within legends). **e** Representative immunofluorescence staining of the P0 cerebellum against Ki67; the regions marked by the white box are magnified in the upper panels. **f** Bar graph shows the average Ki67+ thickness in the P0 cerebellum (*N* = 3 per group). **g** Representative images of immunofluorescence staining for calbindin in the P8 cerebellum. Boxed regions are magnified as indicated. **h** Bar graph shows the average length of the primary dendrite (*N* = 3 per group; 30 PCs per group). **i** *Rbm4* knockout compromised BDNF-TrkB signaling, prolonged GCP cell-cycle, and impaired PC dendritic arborization. Prenatal supplementation with 7,8-DHF reversed *Rbm4*-deficiency caused defects. Scale bars, 200 μm (**c**); 100 μm (**e**, **g**). *P*-values and error bars are the same as in Fig. 1.

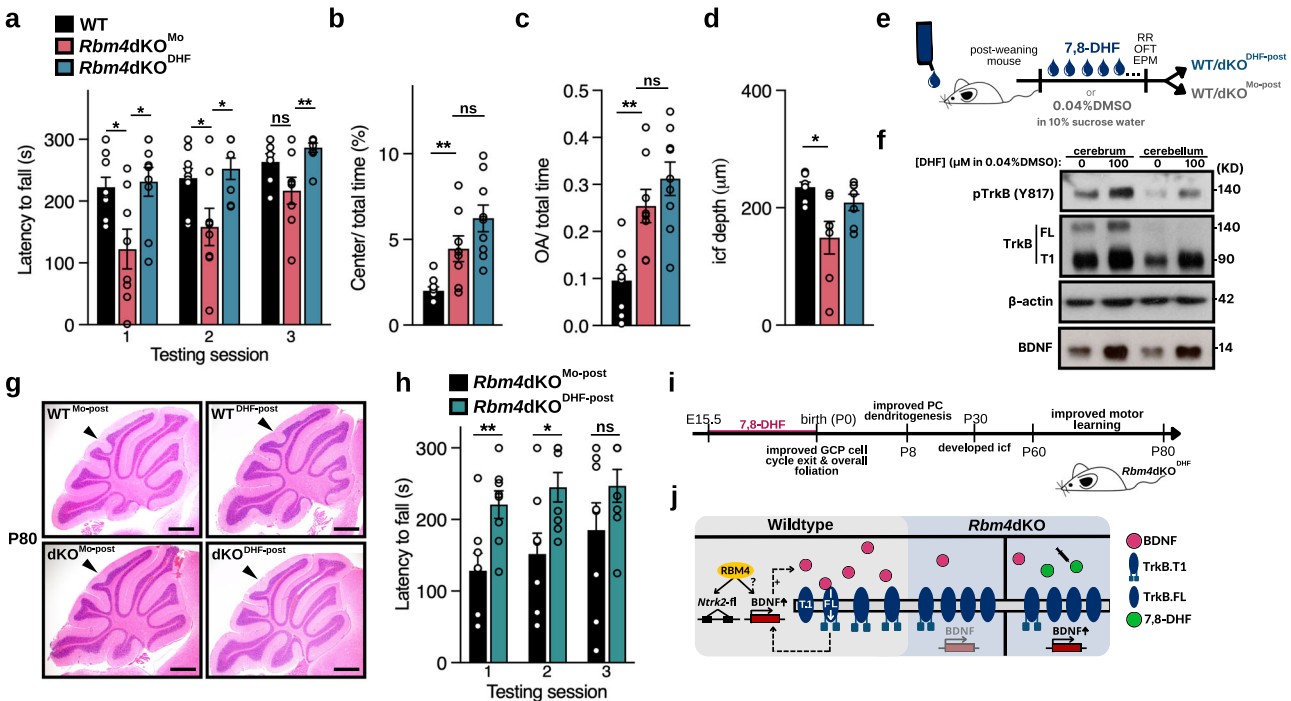

**Fig. 7 Prenatal supplementation with 7,8-DHF improves motor learning of *Rbm4*dKO mice. a** Rotarod test was performed as in Fig. 2a. Bar graph shows latency to fall (s) of each genotype/treatment group in three tests (WT $N = 9$; *Rbm4*dKO$^{Mo}$ $N = 8$; *Rbm4*dKO$^{DHF}$ $N = 9$). **b** Open-field test was performed as in Fig. 2b. Bar graph shows the average cumulative duration spent in the center of the arena, computed as the percentage of total habituation time (60 min). (WT N = 9; *Rbm4*dKO$^{Mo}$ N = 8; *Rbm4*dKO$^{DHF}$ N = 9). **c** EPM test was performed as in Fig. 2c. Bar graph shows the relative duration spent in the open arm during the 5-min habituation (WT $N = 9$; *Rbm4*dKO$^{Mo}$ $N = 8$; *Rbm4*dKO$^{DHF}$ $N = 9$). **d** Average icf depth at P80 ($N = 7$ per group). **e** Schematic of chronic 7,8-DHF administration in post-weaning mice. **f** Representative immunoblotting analysis of the lysates collected from P55 *Rbm4*dKO mice chronically treated with mock or 7,8-DHF drinking solution post-weaning. **g** Representative HE staining of P80 cerebellar vermis. **h** Rotarod analysis of chronically treated animals ($N = 8$ per group). **i** Summary of the developmental course of *Rbm4*dKO$^{DHF}$ mice. Prenatal treatment of *Rbm4*dKO mice with 7,8-DHF improved cerebellar development and motor learning. **j** Molecular mechanisms underlying RBM4-mediated regulation of BDNF-TrkB signaling and the action of 7,8-DHF. RBM4 enhances BDNF-TrkB signaling via an unknown mechanism and increases full-length TrkB by regulating alternative splicing. BDNF activates TrkB signaling and promotes BDNF expression (a positive feedback loop). *Rbm4* knockout attenuates BDNF-TrkB signaling, whereas 7,8-DHF may activate the remaining viable full-length TrkB and restore BDNF-TrkB signaling in *Rbm4*dKO embryos. Scale bars, 500 μm. *P*-values and error bars are the same as in Fig. 1.

malformation in *Rbm4*dKO (Fig. 6). Therefore, we assessed whether prenatal treatment with 7,8-DHF could improve motor learning of adult *Rbm4*dKO mice in the rotarod test. *Rbm4*dKO$^{Mo}$ mice exhibited delayed motor learning, as previously observed with untreated *Rbm4*dKO mice (Fig. 7a; Supplementary Fig. 5a for training). Nevertheless, WT$^{DHF}$ and WT$^{Mo}$ mice showed no significant differences in the rotarod test (Supplementary Fig. 5b). This result indicated that prenatal treatment with 7,8-DHF ameliorated motor learning deficits.

We next assessed the locomotion and anxiety level of *Rbm4*dKO$^{DHF}$ mice. *Rbm4*dKO$^{DHF}$ mice exhibited similar level of hyperactivity and exploration in the center of the open-field arena as that of *Rbm4*dKO$^{Mo}$ (Fig. 7b and Supplementary Fig. 5c). Consistent with this observation, EPM revealed that the anxiety level of *Rbm4*dKO$^{DHF}$ was comparable to that of *Rbm4*dKO$^{Mo}$ (Fig. 7c and Supplementary Fig. 5d). Therefore, prenatal treatment with 7,8-DHF improved motor learning but had no statistically significant effect over the increased anti-anxiety-like behaviors inherent to *Rbm4*dKO. The mice (P80) were sacrificed after these behavioral assays for HE staining of the cerebellar sections. As observed at P30 (Fig. 6c), the icf was partially recovered in P80 *Rbm4*dKO$^{DHF}$. Collectively, 7,8-DHF promoted cerebellar development in the neonatal period, likely contributing to normal morphology and function of the adult cerebellum (Fig. 7d and Supplementary Fig. 5e). The correlation

between the neuroanatomical and behavioral phenotypes is discussed in the next section.

In light of mounting evidence implicating 7,8-DHF in therapeutic intervention for certain neurological disease mouse models with mid-to-late life onset[29,30], we also tested chronic treatment of the agonist in *Rbm4*dKO mice. Post-weaning WT and *Rbm4*dKO mice were provided access to drinking water containing 7,8-DHF or DMSO *ad libitum* (respectively termed WT/*Rbm4*dKO$^{DHF-post}$ and WT/*Rbm4*dKO$^{Mo-post}$; Fig. 7e). Immunoblotting revealed an increased level of phosphorylated TrkB and BDNF in both the cerebrum and cerebellum of *Rbm4*dKO$^{DHF-post}$ mice, verifying 7,8-DHF delivery (Fig. 7f). As expected, chronic 7,8-DHF treatment did not improve icf foliation in *Rbm4*dKO mice (Fig. 7g), but it alleviated their delayed motor learning in rotarod analysis (Fig. 7h; Supplementary Fig. 5f for training). Nevertheless, the same chronic treatment did not exert significant effect on rotarod performance of the wild-type group (Supplementary Fig. 5g), an observation that was likely confounded by a ceiling effect.

Together, prenatal 7,8-DHF supplementation mitigated cerebellar defects of *Rbm4*dKO and improved motor learning without continuous exogenous stimulation after birth (Fig. 7i, j). Chronic 7,8-DHF treatment in post-weaning *Rbm4*dKO adult mice acutely abolished their motor learning delay, as was similarly reported in a mouse model for spinocerebellar ataxia 6 (SCA6)[29].

## Discussion

Our attempt to generate conditional *Rbm4*dKO in the mouse embryonic brain was not successful (see Results). Constitutive knockout of *Rbm4*, however, had no significant effect on the cerebrum but impaired cerebellar icf formation (Fig. 1). This study identified deficient BDNF-TrkB signaling as a major cause of cerebellar malformation inherent to *Rbm4* knockout, which impaired differentiation and migration of GCs—leading to an early transient delay of fissure invagination—and a diminution of dendritic arborization of PCs (Fig. 3). These phenotypes were similar to those of mouse models deficient in *Bdnf*, *TrkB*, *Caps2*, or *Vav*[25,12,15,18], suggesting a role of RBM4 in BDNF biogenesis or signaling. We indeed observed reduced BDNF in *Rbm4*dKO mice and fortuitously found an increased level of TrkB.T1 (Fig. 4). BDNF-TrkB signaling is essential for neurogenesis, neurite outgrowth, and synaptic plasticity[31] (Fig. 6i, Wild type). Although TrkB.T1 harnesses a wide range of cellular activities in neurons and other cell types[32], it nevertheless regulates BDNF-induced TrkB signaling in a dominant-negative manner by blocking BDNF signaling or sequestering BDNF[33,34]. Therefore, *Rbm4* knockout impaired cerebellar development in part by downregulating the BDNF-TrkB pathway (Fig. 6i, *Rbm4*dKO).

To gain a clue as to how RBM4 regulates BDNF expression, we performed RNA-seq on E13.5 wild-type and *Rbm4*dKO brains. *Rbm4* knockout altered the splicing pattern of ~900 genes (~1100 splicing events with False Discovery Rate<0.005; see Materials and Methods for data deposition). Among the transcription factors, *Nsmf* and *Hsf1* have been implicated in BDNF[35,36]. RT-PCR confirmed exon 6 skipping of *Nsmf* and intron 6 retention of *Hsf1* in P7 *Rbm4*dKO GCs (Supplementary Fig. 6a, knockout GC, and diagram). The same result was reproduced by short-hairpin RNA (shRNA)-mediated *Rbm4* knockdown in wild-type GCs (Supplementary Fig. 6a, knockdown GC, and diagram). We hypothesized that *Rbm4* knockout increased the expression of exon 6-truncated NSMF(ΔE6), which lacks the nuclear localization signal, and decreased HSF1, leading to BDNF downregulation. Overexpression of full-length NSMF or HSF1 in RBM4-depleted GCs restored BDNF expression, whereas NSMF(ΔE6) had no effect, as predicted (Supplementary Fig. 6b, c). Therefore, RBM4 likely regulates BDNF expression through splicing regulation of its transactivators (Supplementary Fig. 6d). Although *Ntrk2* was not identified by RNA-seq, our result indicated that RBM4 could regulate alternative selection of terminal exons of *Ntrk2* (Fig. 5). In light of the positive BDNF-TrkB autoregulatory loops, RBM4 may also upregulate *Bdnf* expression via its activity in promoting the expression of full-length TrkB (Fig. 5). Finally, a reduction of *Bdnf* expression in most tissues of *Rbm4*dKO mice suggested that RBM4 regulates BDNF expression systemically via a mechanism similar to that used in the central nervous system (Supplementary Fig. 6e). An in-depth analysis of RNA-seq is underway to determine how RBM4 regulates the BDNF-TrkB pathway.

In contrast to other cerebellar folia, lobules VI and VII receive nonmotor inputs from cerebral association areas and nuclei of the inferior olive and basis pontism, functioning to modulate motor learning, exploratory behavior, and visuospatial ability[8]. As noted above, hypoplasia of lobules VI and VII has been observed in mouse models defective in BDNF signaling. Behavioral analysis has revealed that *Bdnf* or *Caps2* knockout impairs rotarod performance of mice[15,37]. *Ntrk2*-deficient mice display increased exploratory behavior in the open field and EPM, whereas *Bdnf* knockout mice behave differently in locomotion[38,39]. We found that *Rbm4*dKO mice exhibited behavioral phenotypes similar—at least in part—to those of *Bdnf* knockout mice (Fig. 2), suggesting a correlation between BDNF deficiency, hypoplasia of lobules VI and VII, and anomalies in motor learning/locomotion.

BDNF or TrkB agonist has been widely applied to alleviate depression or anxiety-like behaviors[40–42]. 7,8-DHF can improve spatial memory and increase the spine density in a mouse model of Alzheimer's disease[29]. A recent report shows that persistent 7,8-DHF administration improves motor coordination of a SCA6 mouse model[43]. In this study, we primarily characterized *Rbm4*dKO mice prenatally supplemented with 7,8-DHF. Prenatal reactivation of TrkB signaling could restore cerebellar foliation and improve the motor learning as the mice reached adulthood. To better our understanding of how BDNF-TrkB signaling affects cerebellar development and define the effective therapeutic time window, we also tested two other temporal courses of 7,8-DHF intervention regimen. Neonatal and post-weaning administration strategies failed to facilitate timely icf formation, underscoring the importance of prenatal BDNF-TrkB signaling in cerebellar foliation. The result that post-weaning treatment ameliorated motor learning deficits of *Rbm4*dKO mice (Fig. 7), however, implied that BDNF affects icf formation and motor learning through different pathways. It is also possible that behavioral deficits of *Rbm4*dKO mice are in part attributed to BDNF deficiency in other brain regions. Cerebellar cell type-specific knockouts are necessary to more specifically pin down the role of RBM4 in developmental and behavioral deficits associated with the cerebellum in future studies.

Although *RBM4* is not yet known to be linked to any neuro-developmental disorders, our result that chronic treatment with 7,8-DHF improved motor learning in young *Rbm4*dKO adults (Fig. 7h) provides a means to correct behavioral abnormalities associated with BDNF deficiency.

## Methods

**Mice**. All animal care and experimental protocols were reviewed and approved by the Institutional Animal Care and Use Committee (IACUC, protocol IDs 13-04-547 and 19-12-1370) of Academia Sinica and compliant with the Ministry of Science and Technology, Taiwan. Mice were maintained in a 12 h-dark/light cycle at specific pathogen free facility (Experimental Animal Facilities, Academia Sinica), with standard food and water provided *ad libitum*. Eight-week-old male mice were used for the behavioral tests. E18.5 to eleven-week-old mice were used for histological and biochemical analyses. Male and female pups at P7 were used for primary granule cell cultures.

**Constitutive *Rbm4*dKO mice**. *Rbm4a* knockout mice were generated in Lin et al.[2], in which a large portion of exon 2 was replaced by a segment encompassing the enhanced green fluorescent protein (EGFP) gene and a neomycin resistance gene and flanked by two *loxP* sites. The *loxP*-EcoRI sequence was inserted into intron 1 of *Rbm4b* in *Rbm4a* knockout ($Rbm4a^{-/-}$) mice by the CRISPR/Cas9 paired-nicking approach. A paired guide RNA (5'-ctgtgtacctctccacagcgc and 5'-cctccaaaaacccgtaacta) was introduced into *Rbm4a* knockout embryos along with Cas9$^{D10A}$ mRNA and single-stranded oligonucleotide (ssODN) (5'-gttg ggccgctgaggatagggggtccagtgttgggtgaaagagggctttcgttcttggagtatggtgggg aggggataT*cgcgaattc*ATAACTTCGTATAGCATACATTATACG AAGTTAT*ctgtggagaggtacacagatttccctccaaaaacccgtaact*a**ggg**gacattc agatacttcatgccttccacttgtcttt; EcoRI and *loxP* sequences are shown in Italics and capitalized respectively, and modified PAM is underlined; Integrated DNA Technologies). A T7 promoter sequence was added upstream of gRNA sequences, and a partial tracrRNA sequence (5'-GTTTTAGAGCTAGAAATAGC) was added downstream of the gRNA sequence. The oligonucleotide was annealed with reverse tracrRNA (5'- TTTAAAAGCACC GACTCGGTGCCACTTTTTCAAGTTGATAACGGACTAGCC TTATTTTAACTTGCTATTTCTAGCTCTAAAAC) and PCR

amplified using Phusion DNA polymerase (Thermo-Fisher Scientific) according to the manufacturer's instruction. The amplified product was purified with QIAquick PCR purification Kit (Qiagen, 28104) and served as the in vitro transcription template. Single guide RNAs (sgRNAs) were synthesized using HiScribe™ T7 Quick High Yield RNA Synthesis Kit from NEB (New England Biolabs, E2050S). Cas9$^{D10A}$ mRNA was transcribed from the pCAG-T3-hCasD10A-pA plasmid (Addgene, 51638) using the mMESSAGE mMACHINE™T7 ULTRA Transcription Kit (Thermo Fisher Scientific, AM1345), purified using MEGAclear Transcription Clean-up Kit (Thermo-Fisher Scientific, AM1908) and eluted with injection buffer (10 mM Tris-HCl and 0.1 mM EDTA, pH7.2). The quality and quantity of RNAs were analyzed using NanoDrop ND-1000 (Thermo-Fisher Scientific).

For mice production, Cas9$^{D10A}$ mRNA, sgRNAs, and ssODN were diluted with injection buffer to final concentrations of 100, 10 and 100 ng/μl, respectively. Three- to four-week-old of Rbm4a knockout female mice were super-ovulated with 3.75-5 units of pregnant mare gonadotropin serum (PMGS, Sigma-Aldrich) followed by 3.75-5 units of human chorionic gonadotropin (hCG; Sigma-Aldrich) 46 h later. Super-ovulated female mice were set mating to male mice and one-cell stage zygotes were collected on the next day. The mixture of Cas9$^{D10A}$ mRNA, sgRNAs and ssODN was injected into both pronuclei and cytoplasm of the zygotes. Injected zygotes were cultured in KSOM medium in humidified and 5% $CO_2$ incubator at 37 °C overnight. Two-cell stage embryos were transferred into the oviduct of 0.5-dpc pseudo-pregnant ICR female mice.

After genotype confirmation, heterozygous mice (Rbm4a$^{-/-}$; Rbm4b$^{f/+}$) were mated to EIIa-Cre mice (Jax stock no. 003724) to obtain Rbm4a$^{+/-}$;Rbm4b$^{+/-}$ mice. Rbm4 double heterozygous mice were intercrossed to generate wild-type, heterozygotes, and Rbm4dKO; their genotypes were confirmed by PCR.

**Genotyping**. For genotyping, genomic DNA extracted from the skin tissues of embryos or tail biopsy of neonatal pups was subjected to PCR using an Applied Biosystems™2720 real-time PCR instrument (Thermo Fisher Scientific). To detect Rbm4a and Rbm4b, primers complementary to intron 1 and exon 2 of each gene were used. The 568-bp PCR products were subjected to EcoRI digestion, generating the fragments of 202 and 366 bp, indicating successful loxP knockin. PCR of the fragment using primers complementary to EGFP and the respective gene was performed to verify Rbm4a/b knockout. DNA was amplified in a total volume of 20 μl containing 10 μl of Red Dye Genotyping Master Mix (ARROWTEC), 10 μM of each primer, and approximately 100 ng of template DNA. DNA fragments were run on 1.5% agarose gel in 0.5×Tris/acetate/EDTA (TAE) buffer, detected with ethidium-bromide staining, and visualized using Gel Doc™ EZ Imaging System with Image Lab™ Software (Bio-Rad, 1708270; version 5.1). All PCR primers used are listed in Supplementary Table 1.

**RT-PCR and RT-qPCR**. For gene expression analysis, RNA was isolated from mouse embryonic (E13.5) whole brain, or cerebrum or cerebellum at different ages using TRIzol reagent (Thermo Fisher Scientific, 15596018) according to the manufacturer's protocol. Briefly, total RNA was extracted by homogenized tissue lysis in TRIzol reagent followed by phase separation and centrifugation with the addition of chloroform (Merck Millipore, 67-66-3). Following RNA precipitation with 2-propanol (J.T.Baker™, 14-650-208), the RNA pellets were dissolved in autoclaved $H_2O$. In general, 1 μg of total RNA was reverse transcribed into cDNA using oligo (dT) and a SuperScript™III First-strand Synthesis System (Thermo Fisher Scientific, 18080-044). The PCR products

were fractionated by electrophoresis with 1.5% agarose 0.5×TAE gel or 8% PAGE in 0.5×Tris/borate/EDTA (TBE). Quantification of the band intensities was completed with ImageJ (NIH, version 1.52a). Real-time quantitative PCR (RT-qPCR) was performed in triplicates using PerfeCTa qPCR SuperMix (Quantabio, 95050-500) in the LightCycler® 480 Instrument II (Roche Applied Science). Data were normalized by the abundance of Gapdh mRNA. All RT-(q)PCR primers used are listed in Supplementary Table 1.

**Immunoblotting**. For protein extraction, homogenized brain tissues were lysed via re-suspension in radioimmunoprecipitation assay (RIPA) lysis buffer (50 mM Tris-HCl, pH 7.3, 150 mM NaCl, 1% nonidet-P40, 0.1% SDS, 0.5% sodium deoxycholate, protease inhibitors; 1 ml per ~20 mg tissues). Upon centrifugation at 15,000 × g for 10 min at 4 °C, the supernatant was immediately stored at −20 °C until use. The protein concentration was determined using Pierce™ BCA Protein Assay Kit (Thermo Fisher Scientific, 23227) and NanoDrop ND-1000 (Thermo-Fisher Scientific). For immunoblotting, samples were mixed with 2× reducing SDS loading buffer and separated on 8–12% SDS PAGE. The proteins were transferred onto nitrocellulose membrane (Hybond® ECL™, GERPN303D) and probed with specific primary antibodies followed by the corresponding HRP-conjugated secondary antibodies (GE Life Science). For signal visualization, Immobilon ECL Substrate (Millipore, WBKLS0500) was used in conjunction with X-ray film imaging (FUJIFILM, Super RX-N). Quantification was completed with ImageJ (NIH, version 1.52a).

**Antibodies**. Primary antibodies used in this study are as follows: antibodies against RBM4 (1:2000 for immunoblotting, 1:150 for IHC; ProteinTech, 11614-1-AP), Pax6 (1:150; Millipore, AB2237), BrdU (1:50; ABclonal, A1482), Ki67 (1:300; Abcam, ab15580), calbindin (1:200; Abcam, ab108404), NeuN (1:1000; Abcam, ab177487), cleaved caspase-3 (1:100; 5A1E, Cell Signaling Technology, 9664), BDNF (Abcam, ab108319), TrkB (1:1000; Cell Signaling Technology, 4603), phospho-TrkB Tyr816 (1:200; Millipore, ABN1381), phospho-TrkB Tyr817 (1:1500; Invitrogen, MA5-32207), FLAG® (1:2500; Sigma-Aldrich, F1804), GAPDH (1:3000; ProteinTech, 60004-1-Ig), β-actin (1:5000; ProteinTech, 66009-1-Ig) and α-Tubulin (1:3000; Millipore, 05-829). These antibodies were quality-checked by previous publications.

**Histological staining and immunofluorescence**. Paraffin-embedded mouse brain sections were prepared as following. For histology, mice were transcardially perfused with PBS and subsequently with 4% paraformaldehyde in PBS (pH 7.4). Isolated brain or cerebellum was fixed with 4% paraformaldehyde for 24 h, followed by cryoprotection with increasing concentrations of sucrose (from 10 to 30%) in PBS. Brain sections were prepared with a Leica CM3050S cryostat and stored at −20 °C until analysis. Deparaffinized and rehydrated sections were stained with HE or Cresyl Violet (Nissl staining).

For immunohistochemistry, processed sagittal vermis sections were deparaffinized and heat-induced antigen-retrieved in sodium citrate buffer (10 mM trisodium citrate, pH 6.0). The sections were briefly washed with 0.025% Triton X-100 in Tris-buffered saline (TBS; 50 mM Tris-Cl, 150 mM NaCl, pH 7.5) and blocked in TBS with 2% bovine serum albumin (BSA). After blocking, the sections were incubated in primary antibodies diluted in blocking buffer at 4 °C overnight. The slides were then briefly washed with 0.025% Triton X-100 in TBS the following day at room temperature (RT) and peroxidase-quenched by 1.5% hydrogen peroxide in TBS for 15 minutes at RT. Subsequent incubation with the corresponding HRP-conjugated secondary antibody diluted in blocking buffer was performed for 1.5 h at RT.

Finally, substrate-chromogen was developed with 3,3'-diamino-benzidine (DAB, Dako K3468) and counterstained with hematoxylin before mounting. All bright-field images were captured with Olympus BX51 Fluorescence Microscope.

For immunofluorescence, sagittal sections of the cerebellum were subjected to antigen retrieval with sodium citrate buffer followed by quenching endogenous peroxidase activity in 3% hydrogen peroxide. The sections were blocked with 2% BSA in phosphate-buffered saline solution (PBS; 137 mM NaCl, 2.7 mM KCl, 10 mM $Na_2HPO_4$, 1.8 mM $KH_2PO_4$, pH 7.4) and sequentially incubated with specific primary antibodies overnight at 4 °C followed by the corresponding secondary antibodies for 1.5 h at 37 °C. Incubated sections were washed in PBS containing 0.1% Tween 20 (PBST) three times before mounting using ProLong™ Gold Antifade Mountant with DAPI (Thermo Fisher Scientific, P36935). Immunofluorescence images were captured with ZEN desk software (Zeiss) from a Zeiss LSM 780 Confocal Microscope. Morphological analyses were performed on MetaMorph® Image Analysis Software Offline (Molecular Devices).

For immunocytochemistry, sterilized coverslips were placed in 24-well plate and coated with 25 μg/mL poly-D-lysine (Sigma-Aldrich, P6407) prior to plating of freshly isolated cerebellar GCs (see protocol below) overnight in a regular 5% $CO_2$-supplemented incubator at 37 °C overnight. Sample fixation by 4% paraformaldehyde (PFA) in PBS (pH 7.4) was performed the following day (DIV1) for 15 min at RT. After briefly washing in ice-cold PBS, the cells were then permeabilized with 0.25% Triton X-100 in PBS for 10 min at RT. After blocking with 2% BSA in PBST, the cells were briefly washed with PBS and incubated with primary antibodies diluted in blocking buffer at 4 °C overnight. Cells were then washed briefly in PBS and incubated with the respective secondary antibodies. Finally, the nuclei were counterstained and mounted with ProLong™ Gold Antifade Mountant with DAPI (Thermo Fisher Scientific, P36935) for imaging.

**BrdU-pulse labeling**. 5-bromo-2'-deoxyuridine (BrdU)-pulse labeling was performed as Kawamura et al.[44] with minor modifications. P6 mice were intraperitoneally injected with BrdU (10 mg/ml in PBS, Sigma-Aldrich, B5002) at a 50 mg/kg body weight dosage. Whole brains were harvested either 2 or 48 h post-injection and fixed with 4% PFA in PBS overnight at 4 °C.

**Rotarod test**. Rotarod test was performed to assess the motor coordination and balance using Rota-Rod apparatus (47600 Rota-Rod, Ugo Basile, Italy). For the fixed speed rotarod trials, mice were trained to maintain balance on the rod at a constant rotational speed of 4 rpm for 60 sec with a maximum of four trials separated by 10-min inter-trial interval (ITI) before proceeding to the test phase. For the testing, mice were forced to run on a rod accelerating from 4 to 40 rpm in 300 s with an ITI of 15 min. The latency and rpm at which the mice fell off were recorded and analyzed. Two-month-old mice were subjected to rotarod test. Room illumination was maintained at 50 lux, and the apparatus was disinfected with 70% ethanol in between each session.

**Open-field test**. Open-field test was performed in a square arena of 48 × 48 cm made of opaque white acrylic walls to measure the locomotor activity, anxiety, and exploratory behavior by using the Illumination system, Luxmeter, and EthoVision XT Video Tracking System (Noldus). Mouse activity was monitored over 60 min, and the total distance traveled as well as time spent in the center were recorded and analyzed. The central areas were defined as the 16 × 16 cm square located in the center of the arena. Two-month-old mice were individually subjected to the open-field test. Room illumination was maintained at 50 lux, and the apparatus was disinfected with 70% ethanol in between trials.

**Elevated plus maze (EPM)**. The assay was performed essentially as described[30] to assess finer anxiety-related emotional changes in the mouse behaviors. Briefly, each mouse was initially placed at the junction of the plus-shaped apparatus consisted of two open arms (30 × 5 cm) and two close arms (30 × 5 cm; 20 cm wall) elevated 50 cm above the ground. The individual avoidance-approach pattern was video-recorded (EthoVision XT) and analyzed over 5-min duration.

**7,8-DHF supplementation protocol**. For prenatal supplementation, pregnant females were intraperitoneally injected with either 7,8-dihydroxyflavone (7,8-DHF; Sigma-Aldrich, D5446) or vehicle (2% DMSO in PBS) at a daily dosage of 5 mg/kg, starting from embryonic age of E15.5 to birth (P0). The progeny of females that received 7,8-DHF or vehicle are termed WT/$Rbm4$dKO$^{DHF}$ and WT/$Rbm4$dKO$^{Mo}$ upon genotype confirmation, respectively. For neonatal treatment (Mo/DHF-neo), newborn pups were subcutaneously injected with vehicle solution (2% DMSO in PBS) or 7,8-DHF (5 mg/kg body weight) every three days spanning from P5 to P30. For older pups (>P15), injection was humanely administered under mild isoflurane anesthesia if necessary. For chronic supplementation (Mo/DHF-post), post-weaning mice (~>P20) were single-housed under environmental enrichment and given drinking water of vehicle (10% sucrose, 0.04% DMSO) or 7,8-DHF (100 μM in vehicle) ad libitum. On average, the 7,8-DHF-treated animals consumed 20 mL drinking solution daily, which was equivalent to a daily dosage of 508 μg 7,8-DHF. We found no significant nor adverse health problems in either treated groups, and their body weight did not significantly differ from the untreated ones.

**Granule cell isolation, primary culture, AAV transduction, and transfection**. For primary cultures, cerebella isolated from both male and female pups were mixed to exclude any effect of biological sex. Isolation of cerebellar GCs was prepared and modified accordingly from the procedure[45,46]. For GC isolation, a Papain Dissociation System Kit (Worthington, LK003150) was used. Briefly, WT or $Rbm4$dKO mouse was sacrificed at P7, and the cerebellar tissues were tentatively immersed in calcium/magnesium free (CMF)-PBS-EDTA buffer and subsequently lysed in the papain solution followed by incubation in a 37 °C water bath for 30 min. These tissues were then triturated with a sterile fire-polished aerosol pipette pre-coated with Hanks' balanced salt solution (HBSS)-glucose. A single layer of cell suspension was then transferred to a fresh 15-ml Falcon tube with the re-suspension medium (Earle's balanced salt solution [EBSS]+ albumin-ovomucoid inhibitor+ DNase, Worthington) as reconstituted according to the manufacturer's instruction. After centrifugation at ~200 × g for 5 min, the isolated cell pellet was resuspended in 10% FBS medium, filtered with a 70-μm nylon mesh by gravity, and seeded in 24-well plates pre-coated with 25 μg/mL of poly-D-lysine. These cells were nourished in a serum-free culture medium (Gibco™ Neurobasal A medium, Thermo Fisher Scientific, 10888022) supplemented with 2% Gibco™ B27 (Thermo Fisher Scientific, 17504044), 2 mM L-glutamine, and 1% penicillin/streptomycin (Gibco™Thermo Fisher Scientific, 10378016). Smoothened agonist (SAG; Sigma-Aldrich, 566661) was added to a final concentration of 150 nM to sustain and stimulate dendritic outgrowth of the GC network. GCs were maintained in a regular 5% $CO_2$-supplemented incubator at 37 °C overnight. AAV transduction or siRNA-mediated knockdown was performed 24 h after plating for a total duration of 28 or 38 h,

respectively. For siRNA knockdown, either siRNA Negative Control lo GC duplex (Invitrogen, 12935200) or siRBM4 (Invitrogen, sense, GCGUACGCCUUACACCAUGAGUUAU; and antisense, AUAACUCAUGGUGUAAGGCGUACGC, completely complementary to *Rbm4a* and 80% match to *Rbm4b*)[5] was introduced using Lipofectamine™ RNAiMAX Transfection Reagent (Invitrogen, 13778075) according to manufacturer's instruction.

**Mouse E13.5 brain RNA-sequencing.** Total RNA was isolated from E13.5 brains of wild-type and *Rbm4*dKO embryos using TRIzol reagent as described above. After poly(A) selection with an oligo (dT) primer, the RNA samples were treated with DNase I and then subjected to cDNA library construction KAPA mRNA Hyper-Prep Kit (KAPA Biosystems, Roche, Basel, Switzerland) and sequenced using Illumina Novaseq 6000 platform. A total of 40 million 150 bp paired-end reads were generated for each sample. Data was analyzed using rMATS[47] and deposited in the NCBI SRA database with the BioProject accession ID PRJNA980846. Detailed RNA-seq data analysis will be provided in near future studies (Shen et al., manuscript in preparation). To verify splicing changes in Nsmf and HSF1, RT-PCR was performed using total RNA extracted from E13.5 brains or mock or *Rbm4* knockdown GC (see above) using primers (Supplementary Table 1).

**BDNF gene expression assay.** Three pCMV6-Entry based vectors (empty, PS100001; Nsmf, MR221711; Hsf1, MR208087) were obtained from Origene. NSMF(ΔE6) was generated by deleting exon 6 using a PCR-based method; the sequence was verified by Sanger sequencing. *Rbm4* was depleted in P7 GCs at DIV1 by transient transfection of pLKO.1-shRBM4-puro or luciferase as control (pLKO.1-shLuc-puro, RNAi Core Facility, Academia Sinica) using Lipofectamine 2000 (Invitrogen)[4]. For rescue experiments, GCs were co-transfected with the shRBM4 and pCMV6-Entry vectors expressing NSMF or HSF1. RT-PCR and immunoblotting (see "Methods") were subsequently performed to evaluate BDNF expression.

**Statistics and reproducibility.** No statistical methods were used to predetermine sample sizes. For quantification of the cerebellar foliation, the folding index was calculated as described[48]. The Student $t$-test or one-way Analysis of variance (ANOVA) was performed with a minimum of three independent experimental groups to determine the statistical significance between the treatments. The standard deviation (SD) and standard error of the mean (SEM) were calculated using Microsoft Excel and plotted with software GraphPad Prism 9.5.1 (Serial number: GPS-2746004-T###-#####). $P$-value $< 0.05$ was considered statistically significant. All error bars shown are SEM unless otherwise indicated. Sample sizes are indicated in figure legends or otherwise provided in figure panel.

**Reporting summary.** Further information on research design is available in the Nature Portfolio Reporting Summary linked to this article.

## Data availability
Numerical data points underlying all graphs are available in the Supplementary data file. Uncropped images of gels and blots for this study are presented in Supplementary Fig. 7. The RNA sequencing result is deposited and available at NCBI SRA database (BioProject accession ID: PRJNA980846). All other supporting data of this study are available from the corresponding author upon reasonable request.

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

## Acknowledgements

We appreciate Michisuke Yuzaki and Wataru Kakegawa (Keio University School of Medicine, Tokyo, Japan) for discussion and critical reading of the manuscript. We thank Yao-Ming Chang for RNA-seq data analysis, and Yi-Shuian Huang, Hsu-Wen Chao, Pei-Lin Cheng, Ling-Hui Li, Guey-Shin Wang, and Yu-Ting Yen for materials or discussion of the experiments (Academia Sinica, Taipei, Taiwan). We thank the Taiwan Animal Consortium funded by the National Science and Technology Council of Taiwan for technical support in behavioral analyses. We also thank Experimental Animal Facilities, Pathology Core, Common Equipment Core-Confocal Microscopy Core Facility of Institute of Biomedical Sciences and NGS High Throughput Genomics Core at BRCAS of Academia Sinica for their technical assistance, National RNAi Core Facility and AAV Core Facility of Academia Sinica for generating shRNA and recombinant AAV (Grant AS-CFII109-103). This study was supported by the intramural fund of Institute of Biomedical Sciences (Academia Sinica) and Grant 112WIA0110115 from National Science and Technology Council of Taiwan to W.-Y.T.

## Author contributions

Y.-Y.T. performed most experiments (except for Fig. 2a, b; Supplementary Figs. 2, 6), interpreted the data, and wrote the manuscript. C.-L.S. maintained mice and treatment, provided initial characterization of *Rbm4*dKO, and performed plasmid constructions for in vitro experiments (Fig. 5d, RT-PCR; Supplementary Fig. 6). D.D. contributed to Fig. 2a, b, Supplementary Figs. 2, 3d (Golgi staining). C.-Y.T. generated *Rbm4*dKO. W.-Y.T. conceived the study, supervised the experiments, and wrote the manuscript.

## Competing interests

The authors declare no competing interests.
