## [Peer Review File · Communications Biology]

Referee expertise:

Referee #1: BDNF and molecular signaling

Referee #2: developmental biology

Referee #3: BDNF and neurological disorders

Reviewers' comments:

Reviewer #1 (Remarks to the Author):

The authors generated Rbm4 knockout mice and showed that BDNF-TrkB signaling is suppressed, exhibiting phenotypes similar to BDNF KO mice. Experiments have been performed using a variety of approaches, and the experimental results are reliable. However, the readers are most interested in how RNA-binding proteins are involved in BDNF expression. It would also be interesting if an association between the Rbm4 gene mutation and disease is revealed. However, the authors do not address any of these in this paper. The authors should address the mechanism by which Rbm4 regulates the expression of BDNF and TrkB genes at least.

Reviewer #2 (Remarks to the Author):

The authors performed morphological and biochemical assays to test Rbm4 function in the cerebellum of dKO mice and in cultured granule neurons. They found that Rbm4 regulates cerebellar development through the BDNF-TrkB signaling pathway. The authors also found that the phenotypes observed in Rbm4 dKO were rescued by prenatal administration of TrkB agonist, including the levels of proteins or mRNA involved in BDNF-TrkB signaling pathways, and morphological abnormality. Furthermore, prenatal administration of 7,8-DHS restored motor learning in Rbm4 dKO mice. Therefore, the authors concluded that Rbm4 plays a role in cerebellar development and function through the BDNF-TrkB signaling pathway.

The authors performed a significant amount of work to develop interesting hypotheses that Rbm4 regulates cerebellar development through the BDNF-TrkB signaling pathway, which has been described nicely. However, it is questionable for me to connect these findings with cerebellar function including behavioral analyses, due to the use of conventional dKO animals and interpretation of the results (see below, comment #1, 2 and 5). The authors previously have reported that Rbm4 plays role in neuronal migration in cortex (Dhananjaya et al, MCB 2019). Because the authors used a conventional Rbm4dKO, it is hard to interpret the results of behavioral experiments (Figure 2 and 7) and which brain area is involved in the Rbm4dKO phenotype. Since BDNF is also important in the cortex, the author may find that Rbm4 also plays a role in the BDNF-TrkB pathway and Rbm4 deletion causes behavioral deficits in the cortex. In addition, the authors performed behavior assays using the mice treated with a TrkB agonist to examine cerebellar function as one of the key experiments in the study. The administration of the drug by subcutaneous injection is not specific to the cerebellum. To connect cellular abnormality and brain function in vivo, it is important to use conditional knockout mice which lose a target protein at a specific brain region or perform a surgical procedure to deliver a drug into a specific brain region.

<Specific comments>

1. The authors generated dKO mice using EIIa-cre mice. Because Cre expression is driven by a viral promoter, it would be hard to interpret the behavioral results to determine the effects that arise from which brain area and cell type (Figure 2). The author could breed their conditional knockout mouse

line with a mouse line expressing Cre protein in a cerebellar-specific manner, including L7-Cre, Atoh1-Cre or Gabra6-Cre lines to assess cerebellar function of Rbm4.

2. In Figure 2A, the assessment of motor learning is questionable. The Rbm4 deletion phenotype does not seem "motor learning" itself, rather, it appears that the knockout mice may exhibit a problem in dealing with a novel environment, possibly due to hyperactivity. Figure 2B also showed the total distance traveled was higher in the knockout mice. If these mice are habituated to the environment for a longer period before motor learning assays (several sessions or days without acceleration), it may help to reduce stress caused by a novel environment, and the author may examine the ability of these mice in motor learning. Also, what is "total distance traveled" in figure 2C?

3. In Figure 3G, because the signals of immunostaining vary between sections or brains due to technical variability, it is hard to compare "field intensity" by the method the author used without normalization. It would be better if these images were re-analyzed. In addition, it is not clear where the NeuN positive cells localize without DAPI signals. It would be better to label layers as shown in the other figures.

4. I have a technical concern regarding Figure 4A-left, 4C, 5B-left, 5C-left and 5D-left. The quantification of mRNA should be done by qPCR (conventional real time PCR) rather than by quantifying PCR products with a gel because it is difficult to determine whether the PCR reactions were amplified linearly and not saturated. For example, 5B showed reduction of Rbm4 by siRbm4 by both methods, but the degree of knockdown is clearly different (gel: ~10% or less vs qPCR: 30%).

5. In Figure 7, because the authors administered the drug 7,8-DHF by subcutaneously, the effects may not be specific to the cerebellum. Therefore, it is difficult to conclude that the behavioral phenotype was rescued due to morphological changes observed in the cerebellum.

6. In figure 6B-G and Figure 7, what are the effects of 7,8-DHF treatment in wild-type animals? If the treatment increases phospho-TrkB or BDNF protein level or changes cellular morphology or behaviors in wild-type animals, it would be hard to conclude that Rbm4-deletion effects are reversed by 7,8-DHF treatment.

Reviewer #3 (Remarks to the Author):

In this study, the authors observed that constitutive knockout of Rbm4 caused cerebellar vermis hypoplasia in specific lobules through TrkB signaling. This is a very interesting study, but it needs some modifications before it can be published.

1 Under the double knockout of Rbm4, is there any compensatory change of the genes that are similar in the function of Rbm4? Besides, the authors established conditional Rbm4a/Rbm4b double-knockout in the embryonic cortex, whether this has any effect on the function of the hippocampus and cerebellum.

2 The authors found that Rbm4 knockout likely reduced anxiety-like behaviors. But I am worried about the clinical level changes. In other words, how Rbm4 changed in these patients who behaviorally reduced anxiety-like behaviors.

3 Rbm4 knockout prolonged the cell cycle of GCPs within the outer EGL and hence delayed differentiation of newly differentiated GCs and also increased the number of apoptotic cells at P0. Basically, when we want to explore the downstream pathways , proteomics and RNA-seq analysis were needed. At the very least, it can provide researchers with more valuable information.

4 The authors assessed whether prenatal treatment with 7,8-DHF could improve motor learning of adult Rbm4dKO mice in the rotarod test. What the effect of treatment with 7,8-DHF on adult Rbm4dKO mice after birth. That would be meaningful in the translational medicine if it could ameliorate motor learning deficits .

Point-to-Point Response

We thank all reviewers for their constructive comments on our manuscript. According to the reviewers' suggestions, we performed the following major experiments (17 new figure panels in total):

1. 7,8-DHF administration of wild-type embryos as control (Supplementary Fig. 4, 5).
2. neonatal and post-weaning 7,8-DHF treatment (Fig. 7, Supplementary Fig. 5).
3. RNA-seq of *Rbm4*dKO brain (Supplementary Fig. 6).

Moreover, we have adopted alternative quantification methods based on the suggestions (3 panels in Fig. 3G; Supplementary Fig. 3D, 4D). The manuscript has been substantially modified; the major changes are now marked in blue.

These new experiments indicate the following: (1) Prenatal 7,8-DHF administration particularly improved cerebellar development of *Rbm4*dKO mice but did not affect that of the wild-type animals. (2) Postnatal 7,8-DHF supplementation in the newborn *Rbm4*dKO mice failed to rescue the timely formation of cerebellar intercrural fissure. (3) Post-weaning chronic treatment with 7,8-DHF improved the motor learning of *Rbm4*dKO mice. (4) RBM4 modulates alternative splicing of *Bdnf* transactivators, thereby contributing to BDNF expression. We believe the present revision refines the role of RBM4-BDNF pathway in cerebellar foliation and motor learning behaviors.

We hope that the revised manuscript will meet the Reviewers' expectations.

Reviewer #1

The authors generated *Rbm4* knockout mice and showed that BDNF-TrkB signaling is suppressed, exhibiting phenotypes similar to BDNF KO mice. Experiments have been performed using a variety of approaches, and the experimental results are reliable. However, the readers are most interested in how RNA-binding proteins are involved in BDNF expression. It would also be interesting if an association between the *Rbm4* gene mutation and disease is revealed. However, the authors do not address any of these in this paper. The authors should address the mechanism by which *Rbm4* regulates the expression of BDNF and TrkB genes at least.

1. We have performed transcriptomic analysis in the embryonic brain of *Rbm4*dKO and deposited this result in NCBI SRA database (website is detailed in Materials and Methods). Among the target candidates that exhibited altered splicing in *Rbm4*dKO, *Nsmf* and *Hsf1* have been implicated in transcriptional activation of *Bdnf*. We now show that *Rbm4* knockout or depletion caused exon skipping of *Nsmf* and intron retention of *Hsf1*. Overexpression of NSMF (full-length) and HSF1 restored BDNF expression in P7 *Rbm4*dKO cerebellar granule cells (GCs), indicating that RBM4 regulates the expression of BDNF through alternative splicing control of its transactivators. Since characterization of additional RBM4 targets is currently underway, we describe this result in Discussion and show the data in Supplementary Information. Further details of RNA-seq analysis will be included in our next manuscript (Shen CL *et al.*).
2. RBM4 regulates alternative splicing of *Ntrk2* (TrkB) as shown in Figure 5.
3. We have searched three databases for genetic variations of *RBM4*. (1) ClinVar (ncbi.nlm.nih.gov) revealed 15 chromosome deletions of 11q13 (encompassing *Rbm4*) and four *Rbm4* gene mutations. None of the mutations is associated with neurological disorders. (2) OMIM (Online Mendelian Inheritance in Man) search revealed no known gene-phenotype relationship nor *Rbm4* allelic variants associated with human genetic disease. (3) DECIPHER (<https://www.deciphergenomics.org>) revealed 13 copy-number variants covering *RBM4* genes. Since deletions (1.02 to 134 Mb) cover many other genes and generally cause congenital malformation and/or developmental delay, there is no direct evidence to date linking genetic RBM4 variations to neurological disorders.

Reviewer #2

The authors performed morphological and biochemical assays to test Rbm4 function in the cerebellum of dKO mice and in cultured granule neurons. They found that Rbm4 regulates cerebellar development through the BDNF-TrkB signaling pathway. The authors also found that the phenotypes observed in Rbm4 dKO were rescued by prenatal administration of TrkB agonist, including the levels of proteins or mRNA involved in BDNF-TrkB signaling pathways, and morphological abnormality. Furthermore, prenatal administration of 7,8-DHS restored motor learning in Rbm4 dKO mice. Therefore, the authors concluded that Rbm4 plays a role in cerebellar development and function through the BDNF-TrkB signaling pathway.

The authors performed a significant amount of work to develop interesting hypotheses that Rbm4 regulates cerebellar development through the BDNF-TrkB signaling pathway, which has been described nicely. However, it is questionable for me to connect these findings with cerebellar function including behavioral analyses, due to the use of conventional dKO animals and interpretation of the results (see below, comment #1, 2 and 5). The authors previously have reported that Rbm4 plays role in neuronal migration in cortex (Dhananjaya et al, MCB 2019). Because the authors used a conventional Rbm4dKO, it is hard to interpret the results of behavioral experiments (Figure 2 and 7) and which brain area is involved in the Rbm4dKO phenotype. Since BDNF is also important in the cortex, the author may find that Rbm4 also plays a role in the BDNF-TrkB pathway and Rbm4 deletion causes behavioral deficits in the cortex. In addition, the authors performed behavior assays using the mice treated with a TrkB agonist to examine cerebellar function as one of the key experiments in the study. The administration of the drug by subcutaneous injection is not specific to the cerebellum. To connect cellular abnormality and brain function in vivo, it is important to use conditional knockout mice which lose a target protein at a specific brain region or perform a surgical procedure to deliver a drug into a specific brain region.

We thank this reviewer's thoughtful comments. (1) Our previous *in utero* injection of *Rbm4*-targeting shRNA revealed that RBM4 depletion impaired radial glial cell migration in the ventricular zone. This may be a transient effect, as the gross morphology of *Rbm4*dKO cortical lamination was not visibly impaired. (2) As pointed out by the reviewer, *Rbm4* knockout indeed impaired BDNF expression in other brain regions (*e.g.* cerebrum, Supplementary Fig. 4A) and DHF treatment likely had a systemic effect. However, we hope this reviewer agrees with our arguments regarding the advantage of our current approach (please see below for #1).

1. The authors generated dKO mice using EIIa-cre mice. Because Cre expression is driven by a viral promoter, it would be hard to interpret the behavioral results to determine the effects that arise from which brain area and cell type (Figure 2). The author could breed their conditional knockout mouse line with a mouse line expressing Cre protein in a cerebellar-specific manner, including L7-Cre, *Atoh1*-Cre or *Gabra6*-Cre lines to assess cerebellar function of Rbm4.

We agree with this reviewer's comment that cell-type specific knockout would improve our understanding of RBM4's role in BDNF-TrkB signaling and cerebellar development. As stated in the manuscript, we failed to generate *Emx1*-Cre driven knockout. Nevertheless, conventional *Rbm4* double knockout revealed cerebellar deficits without life-threatening phenotypes. Although the failure of neural progenitor cell knockout does not exclude the possibility of

cerebellum-specific knockouts, we attempt to limit the scope of the present study in constitutive KO characterization, as it still offers the following advantages. (1) Conventional KO mouse models may mimic genetic diseases. (2) Conventional KO may eliminate possible cell-type specific compensation or feedforward control in the neuronal circuit, *e.g.* cerebellar Purkinje cell-granule cell connectivity, so we could observe the effect of BDNF deficiency on a gross scale. We will evaluate *Rbm4* deficiency in a cell-type specific manner in the near future.

2. In Figure 2A, the assessment of motor learning is questionable. The *Rbm4* deletion phenotype does not seem "motor learning" itself, rather, it appears that the knockout mice may exhibit a problem in dealing with a novel environment, possibly due to hyperactivity. Figure 2B also showed the total distance traveled was higher in the knockout mice. If these mice are habituated to the environment for a longer period before motor learning assays (several sessions or days without acceleration), it may help to reduce stress caused by a novel environment, and the author may examine the ability of these mice in motor learning. Also, what is "total distance traveled" in figure 2C?

We conducted the rotarod test to assess motor coordination, which has been considered a sensitive assay for detecting cerebellar dysfunction (*J Neurosci Methods*, 189, 180-5; 2010). Prior to rotarod analysis, mice were transported to the behavior testing room for acclimatization at least one hour without any experimental handling. We then performed four consecutive low-constant-speed training sessions prior to the testing phase on the same day (Supplementary Fig. 2, 5), which should have further allowed the mice to adjust to their new environment and minimize environmental stress potentially induced by apparatus positioning. Importantly, only mice that attained full ambulation by the end of the training phase (*i.e.* 60 seconds in training session 4) proceeded for subsequent data acquisition and analysis in the testing sessions. In addition, the fact that *Rbm4*dKO^{DHF} exhibited markedly improved motor learning but still-persistent hyperactivity in the open-field test (OFT) also retrospectively argued for the motor learning delay in rotarod performance as a genuine phenotype inherent to *Rbm4*dKO, not an observation confounded by anxiety-related hyperactivity. In OFT, we employed EthoVision video tracking system to measure the total distance travelled by each mouse subject over 60 minutes of open-field exploration. The average (N=17 per genotype) was reported in Figure 2C.

3. In Figure 3G, because the signals of immunostaining vary between sections or brains due to technical variability, it is hard to compare "field intensity" by the method the author used without normalization. It would be better if these images were re-analyzed. In addition, it is not clear where the NeuN positive cells localize without DAPI signals. It would be better to label layers as shown in the other figures.

We appreciate this Reviewer for bringing our attention to the validity of our fluorescence quantification method. We have now optimized NeuN staining and included DAPI-merged images. We accordingly replaced averaged NeuN+ field intensity with absolute NeuN+ cell counts (Fig. 3G) to exclude technical variance; the cellular layers are now labelled. We further refined quantification method in Supplementary Fig. 4D with normalization to DAPI+ field intensity and replaced P30 PC field-intensity measurements with averaged PC density, calbindin+ molecular layer thickness, and cumulative dendritic length (Supplementary Fig. 3D).

4. I have a technical concern regarding Figure 4A-left, 4C, 5B-left, 5C-left and 5D-left. The quantification of mRNA should be done by qPCR (conventional real time PCR) rather than by quantifying PCR products with a gel because it is difficult to determine whether the PCR reactions were amplified linearly and not saturated. For example, 5B showed reduction of Rbm4 by siRbm4 by both methods, but the degree of knockdown is clearly different (gel: ~10% or less vs qPCR: 30%).

We have performed RT-qPCR for *Bdnf* mRNA expression (Fig. 4A, and 5B-D), but we quantified the level of proteins (4B) and the ratio of *Ntrk2* splice isoforms (4C, D, 5B-D, right) using ImageJ (National Institutes of Health, Bethesda, Maryland, US).

5. In Figure 7, because the authors administered the drug 7,8-DHF by subcutaneously, the effects may not be specific to the cerebellum. Therefore, it is difficult to conclude that the behavioral phenotype was rescued due to morphological changes observed in the cerebellum.

We agree with this reviewer's comment that although there was a correlation between timely icf formation and rotarod performance, 7,8-DHF may have had a systemic effect (affecting brain regions other than the cerebellum). We have now stated this limitation in the last paragraph of the Discussion section.

6. In figure 6B-G and Figure 7, what are the effects of 7,8-DHF treatment in wild-type animals? If the treatment increases phospho-TrkB or BDNF protein level or changes cellular morphology or behaviors in wild-type animals, it would be hard to conclude that Rbm4-deletion effects are reversed by 7,8-DHF treatment.

According to reviewer's suggestion, we have prenatally treated wild-type embryos with 7,8-DHF (Supplementary Fig. 4). No increase in phospho-TrkB was observed, perhaps due to its near-saturated baseline activity (Supplementary Fig. 4A). In addition, we found no change in GC proliferation and apoptosis, cerebellar morphogenesis (Supplementary Fig. 4E-H), and rotarod performance (Supplementary Fig. 5B) in WT^{DHF} mice, as compared to their mock-treated littermates.

Reviewer #3

In this study, the authors observed that constitutive knockout of *Rbm4* caused cerebellar vermis hypoplasia in specific lobules through TrkB signaling. This is a very interesting study, but it needs some modifications before it can be published.

1. Under the double knockout of *Rbm4*, is there any compensatory change of the genes that are similar in the function of *Rbm4*? Besides, the authors established conditional *Rbm4a/Rbm4b* double-knockout in the embryonic cortex, whether this has any effect on the function of the hippocampus and cerebellum.

We clarified that generation of *Emx1*-Cre-driven knockout was not successful (please see Reviewer #2, point 1). Constitutive knockout of *Rbm4a/b*, albeit reducing BDNF (Supplementary Fig. 4J) in the cerebrum, had no significant effect on its gross morphology and lamination (Supplementary Fig. 1).

2. The authors found that *Rbm4* knockout likely reduced anxiety-like behaviors. But I am worried about the clinical level changes. In other words, how *Rbm4* changed in these patients whose behavior reduced anxiety-like behaviors.

Database search did not reveal any RBM4-associated brain disorders (please see Reviewer #1).

3. *Rbm4* knockout prolonged the cell cycle of GCPs within the outer EGL and hence delayed differentiation of newly differentiated GCs and also increased the number of apoptotic cells at P0. Basically, when we want to explore the downstream pathways, proteomics and RNA-seq analysis were needed. At the very least, it can provide researchers with more valuable information.

We have performed RNA-seq and now show that RBM4 regulates BDNF likely through its upstream transactivators (please see Reviewer #1). Because a manuscript for RBM4-mediated splicing regulation in the developing brain is in preparation, we describe the above result in Discussion and show the data in Supplementary Fig. 6. We also detected aberrant splicing of a set of genes involved in neuronal activity in the cerebellar granule cells of *Rbm4dKO* mice, supporting the role of RBM4 in cerebellar development in part via splicing regulation (Shen CL et al., manuscript in preparation).

4. The authors assessed whether prenatal treatment with 7,8-DHF could improve motor learning of adult *Rbm4dKO* mice in the rotarod test. What is the effect of treatment with 7,8-DHF on adult *Rbm4dKO* mice after birth. That would be meaningful in translational medicine if it could ameliorate motor learning deficits.

We appreciate and agree with this reviewer's suggestion that the effective therapeutic timing and duration warrants further investigation. To address this, we provided post-weaning WT and *Rbm4dKO* mice with unlimited access to drinking water of 7,8-DHF (or vehicle solution as control). Consistent with the prenatally treated group, we found increased phospho-TrkB level concomitant with BDNF upregulation in both the cerebrum and cerebellum of the mutant mice (Fig. 7F). Granted, timely icf formation in *Rbm4dKO*^{DHF-post} was not observed (Fig. 7G).

Importantly, chronic DHF supplementation in *Rbm4d*KO, but not wild-type, mice improved their rotarod performance (Fig. 7H). We reasoned that wild-type mice had sufficient BDNF and their baseline performance had reached the peak. Interestingly, in a recent study, such chronic 7,8-DHF treatment at young adult stage (~6-month-old) improved the motor coordination of a spinocerebellar ataxia 6 mouse model (Cook et al. *Sci. Adv.***8**, eabh3260, 2022) likely by activating TrkB-Akt signaling. Nevertheless, we emphasize that prenatal 7,8-DHF treatment mitigated *Rbm4d*KO-induced cerebellar maldevelopment and motor deficits in adult.

Reviewers' comments:

Reviewer #1 (Remarks to the Author):

The authors provided a minimal description of gene expression. The details of the mechanisms regulating gene expression are not yet clear, but may become clearer with continued research by the authors. As for the other parts of the experiment, it is believed that they have been studied in sufficient detail.

Reviewer #2 (Remarks to the Author):

Previously I stated a major concern that it is questionable to connect the findings of BDNF-TrkB pathway with cerebellar function including behavioral analyses, due to the use of conventional dKO animals and interpretation of the results of 7,8-DHF treatment. I am not sure if my concern was clearly resolved in the revised version.

The author's new data clearly showed that postnatal treatment of 7,8-DHF rescued TrkB-BDNF levels (Fig S4I-K) in Rbm4 knockout without affecting icf foliation defects (Fig. S4K). In addition, chronic treatment of 7,8-DHF rescued the motor behavior phenotypes (Fig 7H) and TrkB-BDNF levels (Fig 7F) in Rbm4 knockout without affecting icf foliation defects (Fig. 7G). These results indicate that cerebellar anatomical defects, including icf abnormality, may not be related to motor learning defects in Rbm4 knockout animals.

However, I have the impression that the authors are attempting to connect their findings of cellular and anatomical abnormality in the cerebellum with delayed motor learning in Rbm4 knockout animals. It is likely that their observation of behavioral deficits are due to different brain areas, potentially together with the cerebellum, therefore, some statements in the manuscript seem to be an overstatement.

On the other hand, the authors also discuss the results of 7,8-DHF in the context of BDNF-targeted therapeutic strategy, which seems more reasonable to me.

Reviewer #3 (Remarks to the Author):

The authors generated constitutive homozygous Rbm4a/Rbm4b double-knockout mice and tested RBM4 function in the cerebellum using a variety of approaches. They found that RBM4 plays a role in cerebellar development and function through the BDNF-TrkB signaling pathway and 7,8-DHF, a small-molecule TrkB agonist, partially rescues the developmental defects of the Rbm4 dKO cerebellum and restored motor learning in Rbm4 dKO mice. This is a very interesting study. However, there are still some issues for me. The details are as follows.

1. The authors generated Rbm4 dKO mice using Ella-cre mice. Cre-mediated recombination occurs in a wide range of tissues in Ella-cre mice. Therefore, it would be hard to interpret the phenotypic changes that come from which specific brain area and cell type. The author should test function of RBM4 in cerebellar-specific Rbm4 dKO mice.

2. The authors found RBM4 regulates the BDNF-TrkB pathway which has been associated with defective motor learning and clinical disorders such as autism. My question is whether there are changes of RBM4 in these clinical patients. The clinical relevance needs to be supplemented.

3. The treatment with 7,8-DHF on adult Rbm4 dKO mice after birth increased level of phosphorylated TrkB and BDNF in both the cerebrum and cerebellum. Because BDNF is a secreted protein and also expressed in other tissues (such as heart, lung, skeletal muscle, testis, prostate, and placenta). I am more interested in whether the protein level of BDNF in serum has changed. If the level of BDNF in the

serum increases, it may affect the normal physiological state of other tissues.

Point-to-Point Response

We thank all reviewers for their comments on our manuscript. According to the reviewers' suggestions, we have revised our conclusion regarding BDNF's influence on cerebellar foliation and motor learning behaviors, and evaluated BDNF expression throughout the body (please see Discussion, blue). We hope that this revision will meet the Reviewers' expectations.

Reviewer #1

The authors provided a minimal description of gene expression. The details of the mechanisms regulating gene expression are not yet clear, but may become clearer with continued research by the authors. As for the other parts of the experiment, it is believed that they have been studied in sufficient detail.

We thank this reviewer's comments and understanding of the fact that our current study of RBM4-mediated BDNF regulation needs further investigation.

Reviewer #2

Previously I stated a major concern that it is questionable to connect the findings of BDNF-TrkB pathway with cerebellar function including behavioral analyses, due to the use of conventional dKO animals and interpretation of the results of 7,8-DHF treatment. I am not sure if my concern was clearly resolved in the revised version.

The author's new data clearly showed that postnatal treatment of 7,8-DHF rescued TrkB-BDNF levels (Fig S4I-K) in *Rbm4* knockout without affecting icf foliation defects (Fig. S4K). In addition, chronic treatment of 7,8-DHF rescued the motor behavior phenotypes (Fig 7H) and TrkB-BDNF levels (Fig 7F) in *Rbm4* knockout without affecting icf foliation defects (Fig. 7G). These results indicate that cerebellar anatomical defects, including icf abnormality, may not be related to motor learning defects in *Rbm4* knockout animals.

However, I have the impression that the authors are attempting to connect their findings of cellular and anatomical abnormality in the cerebellum with delayed motor learning in *Rbm4* knockout animals. It is likely that their observation of behavioral deficits are due to different brain areas, potentially together with the cerebellum, therefore, some statements in the manuscript seem to be an overstatement.

On the other hand, the authors also discuss the results of 7,8-DHF in the context of BDNF-targeted therapeutic strategy, which seems more reasonable to me.

We acknowledge that this study has several limitations, and a few aspects have been overstated. We have now modified our interpretation of 7,8-DHF treatment and describe the possibility that BDNF reduction caused foliation defects and abnormal behaviors in *Rbm4*dKO mice in a partially, or even completely, independent manner (lines 376-378). Moreover, delayed motor learning and reduced anxiety may stem from BDNF deficiency in other brain areas (lines 378-379). We have substantially toned down our statements in the Discussion (the last two paragraphs).

Reviewer #3

1. The authors generated *Rbm4* dKO mice using *Ella-cre* mice. Cre-mediated recombination

occurs in a wide range of tissues in Ella-cre mice. Therefore, it would be hard to interpret the phenotypic changes that come from which specific brain area and cell type. The author should test function of RBM4 in cerebellar-specific *Rbm4* dKO mice.

Cerebellar cell type-specific knockout of RBM4 should improve our understanding of cerebellar development. However, we had a difficulty in generating cell type-specific knockout (of two *Rbm4* genes) in the beginning of this study. As described in our previous response, characterization of constitutive KO, however, can mimic genetic diseases, or may eliminate compensation or reciprocal control of BDNF/TrkB signaling amongst different cell types. Therefore, the scope of this study is limited to constitutive KO. We hope that we would be able to generate specific knockouts in cell types of interest.

2.The authors found RBM4 regulates the BDNF-TrkB pathway which has been associated with defective motor learning and clinical disorders such as autism. My question is whether there are changes of RBM4 in these clinical patients. The clinical relevance needs to be supplemented.

We have described in our previous response, a search of several databases (ClinVar, OMIM, and DECIPHER) did not identify any *RBM4* mutations associated with neurological disorders. Additional searches have now been conducted. However, *RBM4* is not listed in SFARI, a database for genes associated with autism spectrum disorder. PubMed search only reveals a potential link between dysregulation of RBM4 and cancer. Although there is no direct link between RBM4 and neurological diseases, our study provides remedial strategies for *Bdnf*-related disorders (lines 382-384). The clinical relevance of RBM4 in disorder/disease still needs future investigation.

3.The treatment with 7,8-DHF on adult *Rbm4* dKO mice after birth increased level of phosphorylated TrkB and BDNF in both the cerebrum and cerebellum. Because BDNF is a secreted protein and also expressed in other tissues (such as heart, lung, skeletal muscle, testis, prostate, and placenta). I am more interested in whether the protein level of BDNF in serum has changed. If the level of BDNF in the serum increases, it may affect the normal physiological state of other tissues.

We thank this reviewer's comment. It is indeed interesting to know whether RBM4 regulates BDNF expression in other tissues. According to *Sci. Rep.* 13:7740 (2023), the level of BDNF in mouse serum is approximately 1/1000 of the level in human serum. In human serum, BDNF is largely released by platelets; whereas in mouse blood, it originates from other tissues such as skeletal muscle. Therefore, we evaluated *Bdnf* expression in several tissues of P7 *Rbm4*dKO mice. We now provide the data in Supplementary Fig. 6E, showing that the level of *Bdnf* was reduced in most tissues examined (except for heart) in addition to the brain in *Rbm4*dKO. This result suggested that the RBM4-BDNF regulation axis exists across the entire body (lines 354-356).